# MicroRNA-122 supports robust innate immunity in hepatocytes by targeting the RTKs/STAT3 signaling pathway

**Hui Xu[†], Shi-Jun Xu[†], Shu-Juan Xie[‡], Yin Zhang[§], Jian-Hua Yang, Wei-Qi Zhang, Man-Ni Zheng, Hui Zhou, Liang-Hu Qu\***

Key Laboratory of Gene Engineering of the Ministry of Education, State Key Laboratory of Biocontrol, School of Life Sciences, Sun Yat-sen University, Guangzhou, China

**\*For correspondence:**
lssqlh@mail.sysu.edu.cn

[†]These authors contributed equally to this work

**Present address:** [‡]Vaccine Research Institute of Sun Yat-sen University, The Third Affiliated Hospital of Sun Yat-sen University, Guangzhou, China; [§]Guangdong Provincial Key Laboratory of Malignant Tumor Epigenetics and Gene Regulation, Research Center of Medicine, Sun Yat-sen Memorial Hospital, Sun Yat-sen University, Guangzhou, China

**Competing interests:** The authors declare that no competing interests exist.

**Abstract** MicroRNA-122 (miR-122) is the most abundant microRNA in hepatocytes and a central player in liver biology and disease. Herein, we report a previously unknown role for miR-122 in hepatocyte intrinsic innate immunity. Restoration of miR-122 levels in hepatoma cells markedly enhanced the activation of interferons (IFNs) in response to a variety of viral nucleic acids or simulations, especially in response to hepatitis C virus RNA and poly (I:C). Mechanistically, miR-122 downregulated the phosphorylation (Tyr705) of STAT3, thereby removing the negative regulation of STAT3 on IFN-signaling. STAT3 represses IFN expression by inhibiting interferon regulatory factor 1 (IRF1), whereas miR-122 targets MERTK, FGFR1 and IGF1R, three receptor tyrosine kinases (RTKs) that directly promote STAT3 phosphorylation. This work identifies a miR-122–RTKs/STAT3–IRF1–IFNs regulatory circuitry, which may play a pivotal role in regulating hepatocyte innate immunity. These findings renewed our knowledge of miR-122's function and have important implications for the treatment of hepatitis viruses.
DOI: https://doi.org/10.7554/eLife.41159.001

## Introduction

Interferon (IFN)-mediated innate immune responses provide a first line of defense against viral infections (*Wack et al., 2015*). These responses are regulated by various factors from the host, pathogen and environment, and these regulatory mechanisms determine the biological outcomes of immune responses and whether pathogens are cleared effectively or go on to cause chronic infection (*Ivashkiv and Donlin, 2014*). MicroRNAs (miRNAs) are a large family of small non-coding RNAs that negatively regulate gene expression by binding to the 3′-untranslated region (UTR) of target mRNAs to induce translation repression or mRNA degradation (*Ambros, 2004*). It has been demonstrated that miRNAs play important roles in various biological processes, including innate immunity (*Mehta and Baltimore, 2016*). Several miRNAs, typified by miR-155 (*O'Connell et al., 2007*; *Wang et al., 2010*), promote innate immune responses whereas others, such as miR-146 (*Hou et al., 2009*; *Taganov et al., 2006*; *Tang et al., 2009*), miR-29 (*Ma et al., 2011*), miR-126 (*Agudo et al., 2014*) and miR-132 (*Lagos et al., 2010*), negatively regulate innate immune responses. To date, the majority of studies have focused on the function of miRNAs in specialized immune cells, whereas the roles of miRNAs in the antiviral innate immunity of non-immune cells are largely unexplored.

Viral infection is a leading cause of liver diseases, and the first line of immune defense against hepatitis viruses is the intrinsic innate immunity within hepatocytes (*Park and Rehermann, 2014*). Among the five human hepatitis viruses (named hepatitis A to E viruses), chronic hepatitis B virus (HBV) and hepatitis C virus (HCV) infections are responsible for the majority of hepatitis, cirrhosis and hepatocellular carcinoma (HCC) (*Arzumanyan et al., 2013*), and thus have been investigated

extensively. HBV behaves as a stealth virus and is nearly invisible to the innate immune response when it infects hepatocytes, probably because it replicates within viral capsids in the cytoplasm and because the generated viral RNAs are capped and polyadenylated, thereby resembling host mRNAs (*Wieland et al., 2004*). By contrast, HCV infection activates innate immune responses, but it employs an elaborate set of mechanisms to evade and block IFN-based antiviral immunity (*Horner and Gale, 2013*). A few of reports have demonstrated an involvement of miRNAs in hepatocyte innate immunity. Interestingly, several miRNAs are employed by hepatitis viruses for their immune evasion. For example, HCV induces miR-208b and miR-499a-5p to suppress IFNL2, IFNL3 and IFNAR1, and thereby represses IFN signaling in HCV-infected hepatocytes (*Jarret et al., 2016*; *McFarland et al., 2014*). HCV also induces miR-373, which directly targets JAK1 and IRF9 (*Mukherjee et al., 2015*), two important components of the Janus kinase-signal transducer and activator of transcription (JAK-STAT) signaling cascade. Although these studies have revealed an interaction between innate immune-related genes and hepatic miRNAs, the roles of miRNAs in regulating hepatocyte innate immunity are far from fully understood.

miR-122 is the predominant miRNA in mature hepatocytes (*Lagos-Quintana et al., 2002*) and plays a critical role in the maintenance of liver homeostasis and hepatocyte function (*Bandiera et al., 2015*). In particular, miR-122 is downregulated or lost in HCC cells (*Coulouarn et al., 2009*; *Kutay et al., 2006*), and both gain-of-function (*Bai et al., 2009*; *Tsai et al., 2009*; *Xu et al., 2010*) and loss-of-function (*Hsu et al., 2012*; *Tsai et al., 2012*) studies have demonstrated that miR-122 serves as a tumor suppressor. However, miR-122 appears to play several dissimilar roles in the infection of hepatitis viruses. In line with its tumor-suppressor role, miR-122 restricts HBV replication and HBV infection in turn downregulates miR-122 expression (*Chen et al., 2011*; *Song et al., 2013*; *Wang et al., 2012*). By contrast, through interaction with the 5′-end of HCV genomic RNA, miR-122 performs an incompletely understood function that is essential for viral replication (*Henke et al., 2008*; *Jopling et al., 2005*). Accordingly, antagonizing miR-122 as a potential HCV therapeutic strategy (*Lanford et al., 2010*) is currently under phase II clinical studies (*Janssen et al., 2013*). However, analysis of miR-122 expression in liver biopsies from subjects with chronic hepatitis C undergoing IFN therapy showed that reduced miR-122 levels were associated with poorer treatment outcome (*Sarasin-Filipowicz et al., 2009*; *Urban et al., 2010*), suggesting a positive role of miR-122 in HCV therapy. A handful of reports have suggested that miR-122 may regulate IFN signaling by inhibiting suppressor of cytokine signaling 3 (SOCS3) or SOCS1, but their conclusions are diametrically opposed (*Gao et al., 2015*; *Li et al., 2013a*; *Yoshikawa et al., 2012*). Therefore, the role of miR-122 in hepatocyte innate immunity is currently unclear.

As miR-122 is a bona-fide tumor suppressor, we hypothesized that miR-122 might play a positive role in hepatocyte innate immunity. This hypothesis was supported by the previous observations that the innate immune response is robust in primary hepatocytes but strongly impaired in hepatoma cells (*Li et al., 2005a*; *Thomas and Liang, 2016*). In the present work, we systematically investigated the role of miR-122 and its molecular mechanism in regulating antiviral IFN responses. We provide the first evidence that miR-122 promotes antiviral innate immunity by targeting RTKs/STAT3 signaling, which plays a leading role in inflammation and immunity. In addition, we have also dissected the mechanism through which STAT3 represses the IFN response, an important topic in innate immunity. These findings renewed our knowledge about miR-122's role in viral infection and immunity, which has important implications for the treatment of hepatitis viruses.

## Results

### Restoring miR-122 in hepatoma cells enhances antiviral IFN responses

To select a suitable cell model for our study, we examined the functional innate immunity of three human hepatoma cell lines, HepG2, Huh7 and Huh7.5.1 (*Zhong et al., 2005*), in response to JFH1 (*Wakita et al., 2005*) HCV RNA stimulation. By measuring the induction of phosphorylated STAT1 (p-STAT1, a marker of IFN signaling activation), melanoma differentiation-associated protein 5 (MDA5, an interferon-stimulated gene (ISG) that is markedly induced upon IFN signaling activation), and IFN mRNAs (IFN-β, type I; IL-29 and IL-28, type III), we found that HepG2 has a relatively strong innate immunity compared to that of Huh7 and Huh7.5.1 (*Figure 1—figure supplement 1A,B*, *Figure 1—source data 1*), and may be relatively close to the primary human hepatocytes in innate

immune response (*Israelow et al., 2014*; *Thomas et al., 2012*). Therefore, we selected HepG2 for further studies.

Next, we compared the IFN responses of HepG2 cells to different viral nucleic acids or simulants. In addition to HCV RNA (JFH1 and SGR-JFH1), double-stranded RNA (dsRNA) mimetic poly(I:C) and 5′ triphosphate hairpin RNA (3p-hpRNA) also induced strong IFN responses, specifically of type III IFNs (IL-29 and IL-28) (*Figure 1A*). By contrast, transfection with RNAs extracted from HepG2 cells harbouring transient (HBV1.3) or stable (HBV2.15) HBV replicons (*Figure 1—figure supplement 1C*, *Figure 1—source data 2*), in vitro transcribed HBV εRNA (*Sato et al., 2015*) (a structure in pre-genomic HBV RNA that has been shown to induce type III IFNs), viral DNA motif derived from the herpes simplex virus (HSV-60), or U-rich single-stranded RNA derived from HIV (ssRNA40) only induced weak IFN activation (*Figure 1A*, *Figure 1—source data 3*).

To assess the effect of miR-122 on innate immunity, we overexpressed miR-122 in HepG2 cells, and then treated cells with different stimuli. miR-122 mimic transfection in HepG2 cells restored miR-122 expression to a level close to that of normal human liver (*Figure 1B*, *Figure 1—source data 4*). Surprisingly, miR-122 overexpression markedly increased the IFN induction in response to almost all of the stimuli we tested, but did not impact the IFN expression in cells without stimulation (*Figure 1C–F*, *Figure 1—figure supplement 1D*, *Figure 1—source data 5*). Interestingly, when cells were treated with strong IFN agonists (HCV RNA, poly(I:C) and 3p-hpRNA), both type I and type III IFNs were strikingly increased, whereas when cells were treated with weak agonists (HBV1.3 RNA, εRNA and HSV-60), only INF-β was significantly increased. These results might due to the basal expression of type III IFNs, which was very low in cells treated with weak agonists. Notably, within all groups, the activating effect of miR-122 on IFN induction was more remarkable in cells treated with HCV RNA (both JFH1 and SGR-JFH1) or poly(I:C) (*Figure 1C*, *Figure 1—figure supplement 1D*), both of which can form dsRNA structures (*Saito et al., 2008*). ELISA assays showed that both type I and type III IFN proteins were increased by miR-122 overexpression in these two groups (*Figure 1E, F*, *Figure 1—source data 6*). In addition, ISGs downstream of the JAK-STAT cascade, including MDA5, retinoic acid inducible gene I (RIG-I), and the chemokines CXCL9 and CXCL10, were also greatly increased (*Figure 1—figure supplement 1E*, *Figure 1—source data 7*), suggesting that the entire IFN pathway was upregulated. We also performed assays in Huh7 cells. Although not highly significant, miR-122 increased the induction of IFNs and p-STAT1 upon HCV RNA stimulation (*Figure 1—figure supplement 1F*, *Figure 1—source data 8*).

## miR-122 enhances the IFN response to HCV independent of its interaction with HCV

Considering that miR-122 can bind to the HCV 5′ UTR and enhance viral replication, we were interested in whether the remarkable IFN induction in miR-122/HCV RNA-treated cells was caused by an increase in HCV RNA abundance. However, the HCV RNA level was only slightly higher (~1.5 fold increase) in miR-122-treated cells than in control-treated cells (*Figure 2A*, *Figure 2—source data 1*), even though HCV translation was increased by more than 20-fold (*Figure 2—figure supplement 1A,B*, *Figure 2—source data 2*). Analysis of the effect of HCV RNA dosage on IFN activation showed that when the transfection dose rose from 1 μg to 3 μg per ml, the intracellular viral RNA increased 6-fold, but the expression of IFNs only doubled (*Figure 2B*, *Figure 2—source data 3*). Therefore, the effect of miR-122 on IFN induction was not due to the increase in HCV RNA.

To determine whether the miR-122-induced promotion of IFN activation depends on miR-122 binding to the HCV 5′ UTR, we studied a viral RNA mutant (JFH1-M) with single base substitutions in S1 and S2 (*Figure 2C*) that was reported to ablate miR-122 binding (*Jangra et al., 2010*; *Li et al., 2013b*). As expected, mutation of miR-122 binding sites evidently impaired the promoting effect of miR-122 on HCV protein translation, as indicated by the expression of core and non-structural 3 (NS3) proteins (*Figure 2D*, *Figure 2—figure supplement 1C*). By contrast, the mutation did not reduce the activating effect of miR-122 on p-STAT1 (*Figure 2D*) and IFNs (*Figure 2E*, *Figure 2—source data 4*), indicating that binding to HCV RNA is not required for the activating effect of miR-122 on IFNs.

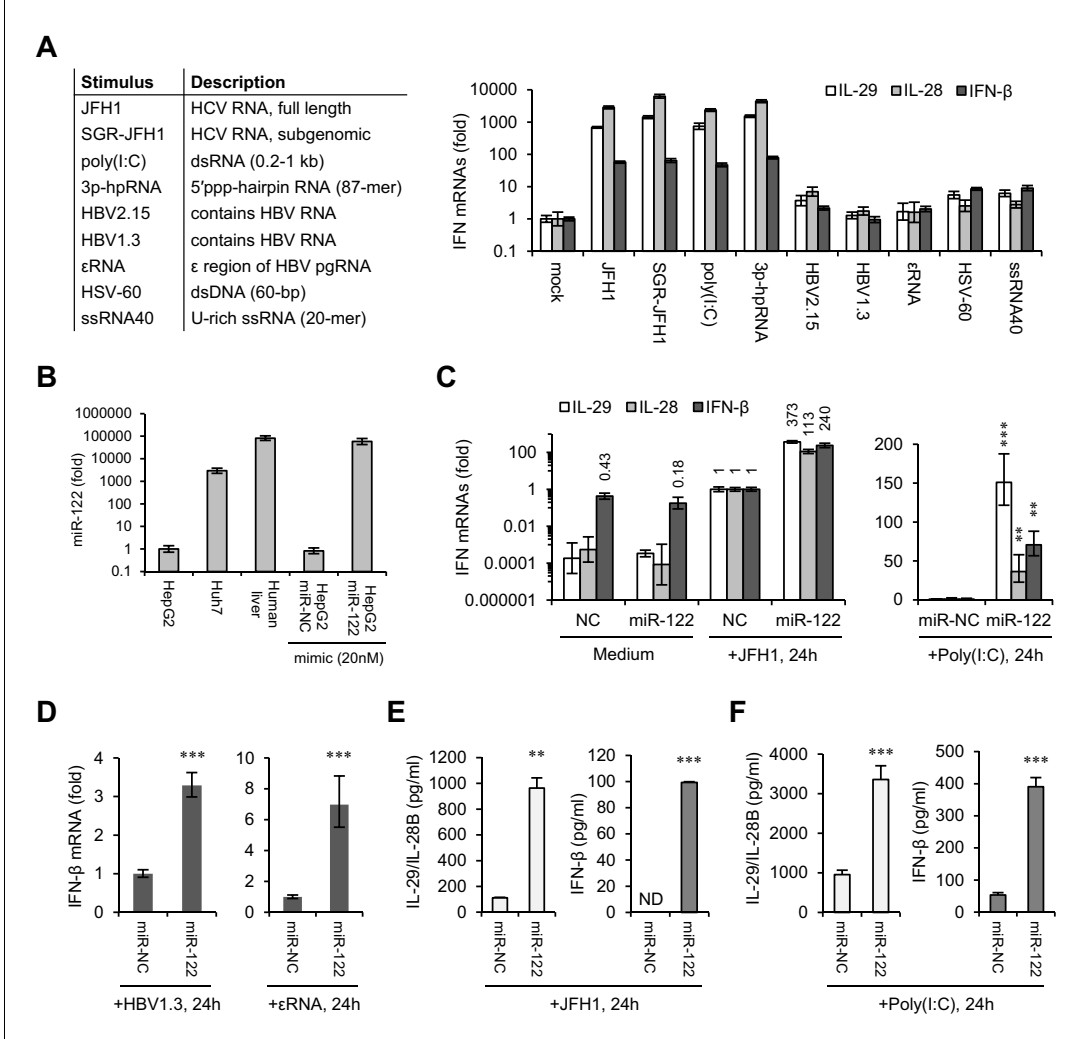

**Figure 1.** miR-122 enhances IFN responses to various viral nucleic acids. (**A**) qRT-PCR analysis of type III (IL-29 and IL-28) and type I (IFN-β) IFN mRNAs in HepG2 cells transfected with the indicated nucleic acids for 24 hr. Except for HSV-60 and ssRNA40, which were transfected at 5 μg/ml, all other RNAs were transfected at 1 μg/ml. The expression of IFNs was normalized to GAPDH and then compared with the levels in cells without stimulation (mock). (**B**) qRT-PCR analysis of miR-122 expression in hepatoma cell lines and in normal liver tissue, as well as of HepG2 cells transfected with miR-122 or negative control (miR-NC) mimics. miR-122 expression was normalized to U6 and then compared with the levels in HepG2 cells. (**C, D**) qRT-PCR analysis of IFN mRNAs in HepG2 cells first treated with miR-122 or control mimics for 2 days, and then transfected with the indicated nucleic acids for 24 hr. Opti-MEM (medium) was used as the negative control. (**E, F**) ELISA analysis of IL-29/IL-28B and IFN-β proteins in HepG2 cells treated as in panel C. IL-29 and IL-28B were detected by the same set of antibodies. N.D., lower than the minimum concentration (3.9 pg/ml) that can be accurately detected. qRT-PCR data are from one experiment that was representative of three independent experiments (mean ±SEM of technical triplicates). ELISA data are from two experiments (mean +SD). *p<0.05, **p<0.01 and ***p<0.001.
DOI: https://doi.org/10.7554/eLife.41159.002

The following source data and figure supplement are available for figure 1:

**Source data 1.** qRT-PCR analysis of IFN expression in HepG2 and Huh7 cells transfected with JFH1 RNA.
DOI: https://doi.org/10.7554/eLife.41159.004

**Source data 2.** qRT-PCR analysis of HBV pgRNA levels in indicated samples.
DOI: https://doi.org/10.7554/eLife.41159.005

**Source data 3.** qRT-PCR analysis of IFN mRNAs in HepG2 cells transfected with different nucleic acids.
DOI: https://doi.org/10.7554/eLife.41159.006

**Source data 4.** qRT-PCR analysis of miR-122 expression in the indicated samples.
DOI: https://doi.org/10.7554/eLife.41159.007

**Source data 5.** qRT-PCR analysis of IFN mRNAs in HepG2 cells transfected with miR-122 and then treated with different nucleic acids.
DOI: https://doi.org/10.7554/eLife.41159.008

*Figure 1 continued on next page*

*Figure 1 continued*

**Source data 6.** ELISA analysis of IFNs in HepG2 cells transfected with miR-122 and then treated with different nucleic acids.
DOI: https://doi.org/10.7554/eLife.41159.009

**Source data 7.** qRT-PCR analysis of ISGs in HepG2 cells transfected with miR-122 and then treated with JFH1.
DOI: https://doi.org/10.7554/eLife.41159.010

**Source data 8.** Analysis of the IFN mRNAs in Huh7 cells transfected with miR-122 and then treated with JFH1.
DOI: https://doi.org/10.7554/eLife.41159.011

**Figure supplement 1.** miR-122 promotes antiviral IFN response.
DOI: https://doi.org/10.7554/eLife.41159.003

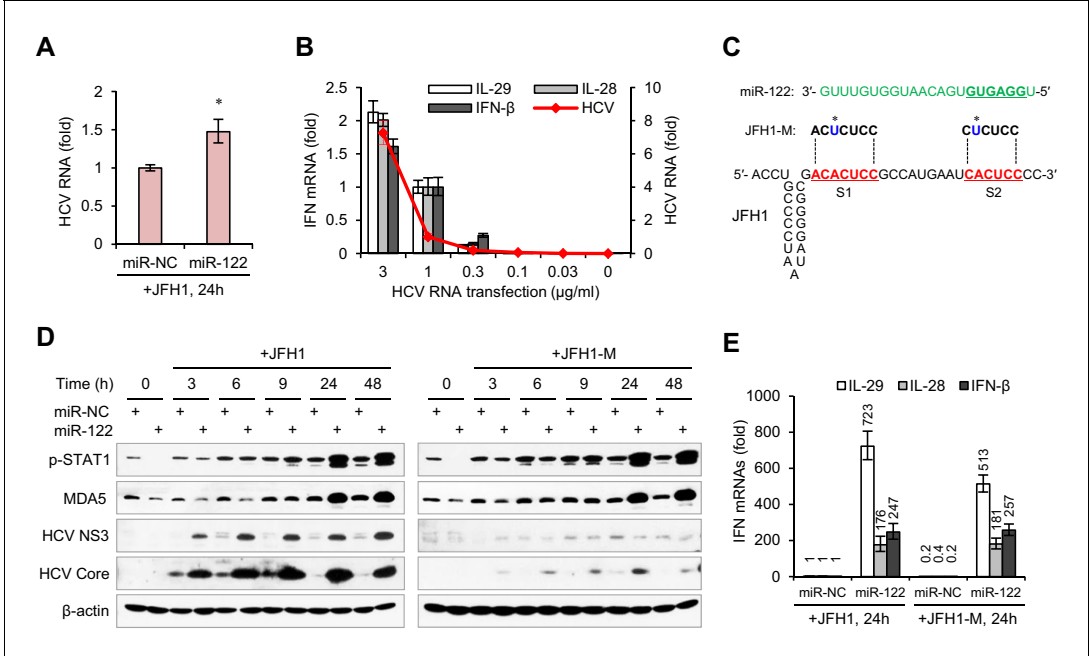

**Figure 2.** miR-122-induced promotion of IFN activation does not depend on miR-122 binding to the HCV 5′ UTR. (**A**) qRT-PCR analysis of the relative level of HCV RNA in HepG2 cells first treated with miR-122 or control mimics for 2 days, and then transfected with JFH1 for 24 hr. (**B**) qRT-PCR analysis of HCV RNA and IFN mRNAs in HepG2 cells transfected with different doses of JFH1 RNA. The expression of IFNs and HCV RNA was normalized to GAPDH and then compared with the levels in cells transfected with 1 µg/ml JFH1 RNA. (**C**) Alignment of miR-122 to the binding sites (S1 and S2) in the 5′ -end of JFH1 and mutant JFH1 (JFH1-M). (**D**) Western blot analysis of p-STAT1 (Tyr701), MDA5, HCV Core and NS3 in HepG2 cells first treated with miR-122 or negative control (miR-NC) mimics for 2 days, and then transfected with JFH1 RNA for 3–48 hr. (**E**) qRT-PCR comparison of IFN expression in HepG2 cells treated with JFH1 or JFH1-M. qRT-PCR data are from one experiment that was representative of three experiments (mean ±SEM of technical triplicates). *p<0.05.
DOI: https://doi.org/10.7554/eLife.41159.012

The following source data and figure supplement are available for figure 2:

**Source data 1.** qRT-PCR analysis of HCV RNA in HepG2 cells.
DOI: https://doi.org/10.7554/eLife.41159.014

**Source data 2.** Luciferase assays of the Gluc reporter treated with miR-122 mimic or XRN1 siRNA.
DOI: https://doi.org/10.7554/eLife.41159.015

**Source data 3.** qRT-PCR analysis of HCV RNA and IFN mRNAs in HepG2 cells transfected with different doses of JFH1 RNA.
DOI: https://doi.org/10.7554/eLife.41159.016

**Source data 4.** qRT-PCR comparison of IFN expression in HepG2 cells treated with JFH1 or JFH1-M.
DOI: https://doi.org/10.7554/eLife.41159.017

**Figure supplement 1.** The effect of miR-122 on HCV translation.
DOI: https://doi.org/10.7554/eLife.41159.013

## miR-122 promotes the IFN response by suppressing STAT3 phosphorylation

Several reports have suggested that miR-122 may regulate IFN signaling by suppressing the negative regulators of the JAK-STAT cascade, such as SOCS3 and SOCS1 (*Gao et al., 2015*; *Li et al., 2013a*; *Yoshikawa et al., 2012*), which can bind to JAKs and inhibit the phosphorylation of STAT by JAK kinases. However, we found that miR-122 did not promote the induction of p-STAT1 in HepG2 cells treated with either IFN-β or IL-29 (*Figure 3—figure supplement 1A*), suggesting that the effect of miR-122 on IFN signaling was not achieved by the suppression of these JAK inhibitors. In line with this hypothesis, miR-122 did not inhibit the mRNA expression of five tested SOCS genes (SOCS1–5) (*Figure 3—figure supplement 1B*, *Figure 3—source data 1*).

When we investigated the effects of miR-122 on several immunity-related signaling pathways using the Multi-Pathway Reporter Arrays, we found that miR-122 significantly repressed the activity of STAT3 (*Figure 3A*, *Figure 3—source data 2*), a known negative regulator of type I IFN-mediated antiviral responses (*Ho and Ivashkiv, 2006*; *Wang et al., 2011*). So, we first tested whether miR-122 downregulates the expression of STAT3. Intriguingly, miR-122 overexpression strongly inhibited tyrosine (Tyr705) phosphorylation of STAT3 (p-STAT3) but did not affect the total STAT3 protein level, independent of HCV RNA or poly(I:C) stimulation (*Figure 3B*). Although miR-122 appeared to downregulate STAT3 mRNA expression slightly, this alteration was extremely mild (*Figure 3—figure supplement 1C*, *Figure 3—source data 3*). Moreover, comparing the effects of miR-122 overexpression and small interfering RNA (siRNA)-mediated STAT3 knockdown confirmed the specific regulatory effect of miR-122 on p-STAT3 (*Figure 3C*, *Figure 3—figure supplement 1D*).

Excitingly, STAT3 knockdown greatly increased IFN production (*Figure 3D–F*, *Figure 3—source datas 4–6*) and p-STAT1 induction (*Figure 3—figure supplement 1E*) in response to HCV RNA or poly(I:C) stimulation. Notably, the IFN level was much higher in the STAT3 knockdown cells than in the miR-122-treated cells (*Figure 3D*). Furthermore, blocking of STAT3 phosphorylation using the chemical inhibitor S3I-201 (*Siddiquee et al., 2007*) or the natural compound cryptotanshinone (CTS) (*Shin et al., 2009*) also increased the IFN response (*Figure 3G,H*, *Figure 3—figure supplement 1F*, *Figure 3—source data 7*). Therefore, these data demonstrate that miR-122 regulates IFN activation by repressing STAT3 phosphorylation.

We also performed assays in Huh7 cells. Unexpectedly, although miR-122 promoted the IFN response, neither overexpression nor blocking of miR-122 affected the expression of p-STAT3 in Huh7 cells (*Figure 3—figure supplement 2A and B*, *Figure 3—source data 8*). Nevertheless, knockdown of STAT3 increased the IFN activation in Huh7 cells. To investigate whether miR-122 represses p-STAT3 and promotes IFN activation in other liver cells, we employed Hep3B and BNL CL.2. (The former is derived from human HCC cells and the latter from 'normal' mouse liver cells (immortalized).) Although the p-STAT3 level is relative low in these two cell lines, overexpression of miR-122 obviously reduced p-STAT3 (*Figure 3—figure supplement 2C*). Notably, miR-122 overexpression and STAT3 knockdown resulted in very similar upregulation of the IFN activation in Hep3B cells (*Figure 3—figure supplement 2D*, *Figure 3—source data 9*). These results suggest that miR-122 plays similar roles in these different cells, but that the regulatory network from miR-122 to p-STAT3 might be disrupted in Huh7.

## STAT3 inhibits IRF1 expression

To understand how STAT3 regulates IFN activation, we analysed the expression of five transcription factors that are responsible for IFN activation: IRF1, IRF3, IRF7, RELA and NFKB1. Both miR-122 overexpression and STAT3 knockdown substantially increased IRF1 expression at 24 hr post HCV RNA transfection, but did not significantly impact the expression of the other transcription factors (data on IRF7 is not shown because it was barely detected) (*Figure 4A*, *Figure 4—source data 1*). Analysis of the protein expression of IRF1 and IRF3 at different time after HCV RNA stimulation showed that STAT3 knockdown increased IRF1 activation very early or even in unstimulated cells (*Figure 4B*), suggesting that IRF1 upregulation might account for the increased IFN activation.

As expected, IRF1 overexpression was able to trigger IFN expression (*Figure 4C*, *Figure 4—source data 2*) and STAT1 phosphorylation (*Figure 4D*) directly in the absence of stimulus. Consistent with the report showing that IRF1 is specifically required for IFN-λ activation (*Odendall et al., 2014*), IRF1 overexpression primarily induced IFN-λ expression (*Figure 4C*). Notably, IRF1 induced

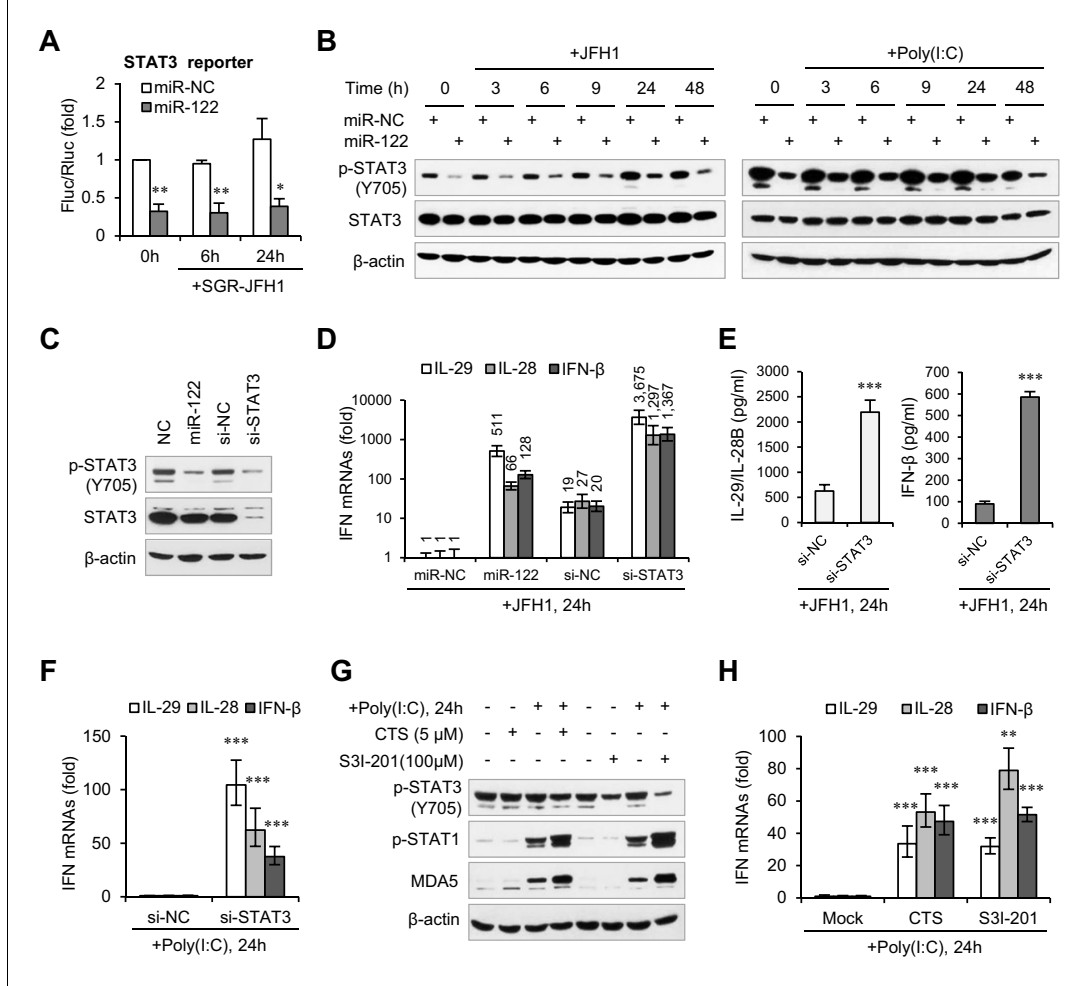

**Figure 3.** miR-122 enhances IFN response by suppressing p-STAT3. (**A**) Luciferase activity of a STAT3-responsible promoter construct in HepG2 cells co-transfected with mimics (NC or miR-122) for 2 days, and then transfected with SGR-JFH1 RNA for the indicated time. (**B**) Western blot analysis of total and phosphorylated STAT3 (p-STAT3, Tyr705) in HepG2 cells first transfected with mimics (NC or miR-122) for 2 days, and then treated with JFH1 RNA or poly(I:C). (**C**) Analysis of STAT3 protein in HepG2 cells treated with mimics (NC or miR-122) or siRNAs (NC or STAT3). (**D, E**) Analysis of the mRNAs (**D**) and proteins (**E**) of IFNs in HepG2 cells treated with siRNAs (NC or STAT3) and then JFH1 RNA. Cells treated with miR-122 or NC mimics were used as controls in panel D. (**F**) Analysis of IFN mRNAs in HepG2 cells treated with siRNAs and then with poly(I:C). (**G, H**) Analysis of p-STAT1 protein and IFN mRNAs in HepG2 cells treated with either S3I-201 or cryptotanshinone (CST) for 24 hr, and then transfected with poly(I:C). Luciferase data are from three experiments (mean +SD). ELISA data are from two experiments (mean +SD). qRT-PCR data are from one experiment that was representative of three experiments (mean ± SEM of technical triplicates). *p < 0.05, **p < 0.01 and ***p < 0.001.

DOI: https://doi.org/10.7554/eLife.41159.018

The following source data and figure supplements are available for figure 3:

**Source data 1.** qRT-PCR analysis of the five SOCS genes in HepG2 cells.
DOI: https://doi.org/10.7554/eLife.41159.021
**Source data 2.** Luciferase activity of a STAT3-responsible promoter construct in HepG2 cells.
DOI: https://doi.org/10.7554/eLife.41159.022
**Source data 3.** qRT-PCR analysis of STAT3 mRNA in HepG2 cells.
DOI: https://doi.org/10.7554/eLife.41159.023
**Source data 4.** qRT-PCR analysis of IFN mRNAs in HepG2 cells treated with siRNAs and then treated with JFH1.
DOI: https://doi.org/10.7554/eLife.41159.024
**Source data 5.** ELISA analysis of IFN proteins in HepG2 cells treated with siRNAs and then treated with JFH1.
DOI: https://doi.org/10.7554/eLife.41159.025
**Source data 6.** qRT-PCR analysis of IFN mRNAs in HepG2 cells treated with siRNAs and then treated with poly(I:C).
DOI: https://doi.org/10.7554/eLife.41159.026
**Source data 7.** qRT-PCR analysis of IFN mRNAs in HepG2 cells treated with either S3I-201 or cryptotanshinone (CST).

*Figure 3 continued on next page*

*Figure 3 continued*

DOI: https://doi.org/10.7554/eLife.41159.027
**Source data 8.** qRT-PCR analysis of IFN mRNAs in Huh7 cells.
DOI: https://doi.org/10.7554/eLife.41159.028
**Source data 9.** qRT-PCR analysis of IFN mRNAs in Hep3B cells.
DOI: https://doi.org/10.7554/eLife.41159.029
**Figure supplement 1.** miR-122 regulates IFN response by repressing STAT3 phosphorylation.
DOI: https://doi.org/10.7554/eLife.41159.019
**Figure supplement 2.** The effects of miR-122 on p-STAT3 and IFN response in different liver cell lines.
DOI: https://doi.org/10.7554/eLife.41159.020

both STAT1 phosphorylation and MDA5 expression in a dose-dependent manner, and even a small amount of IRF1 evidently increased p-STAT1 and MDA5 expression (*Figure 4E*). Comparing IRF1 activity to that of six known transcriptional regulators of IFN-β (*Panne et al., 2007*) also revealed that IRF1 is the major inducer of p-STAT1 (*Figure 4F*). A previous study suggested that STAT3 sequesters STAT1 into STAT1:STAT3 heterodimers and thus inhibits IFN-induced STAT1-dependent IRF1 activation (*Ho and Ivashkiv, 2006*). In our system, however, STAT3 knockdown did not increase the induction of IRF1 by either IFN-β or IL-29 (*Figure 4G*), suggesting that STAT3 might inhibit IRF1 through another mechanism.

## STAT3 binds directly to the promoter and enhancers of IRF1

To investigate whether STAT3 is able to inhibit IRF1 transcriptional activation directly, we analysed the chromatin immunoprecipitation sequencing (ChIP-seq) data available in the UCSC Genome Browser. We noticed that seven STAT3 binding clusters (BS1 to BS7) have been identified on the IRF1 gene locus, and that three clusters (BS1, BS3 and BS4) possess conserved STAT3-binding motifs (*Figure 5A*, *Figure 5—figure supplement 1A*). Because these ChIP-seq data were obtained in cell lines other than hepatocytes, we performed ChIP experiments in HepG2 cells. Interestingly, STAT3 strongly bound to BS1, BS2, BS3 and BS4 sites, whereas p-STAT3 bound to BS2, BS3 and BS4 but not BS1 (*Figure 5B*). By comparison, RELA, an activator of IRF1 (*Figure 4F*), bound only weakly to BS1, BS2 and BS4 (*Figure 5B*). These data indicate that both phosphorylated and un-phosphorylated STAT3 participate in the regulation of IRF1.

To determine which sites are critical for IRF1 regulation, we generated a series of luciferase constructs (*Figure 5A,C*) and assessed their activities. Consistent with previous studies demonstrating that the IRF1 promoter is located immediately upstream of the IRF1 transcriptional start site (TSS1 in *Figure 5A*) (*Harada et al., 1994*), the activities of the P1 and P7 constructs were substantially greater than those of all of the other constructs (*Figure 5C*, *Figure 5—source data 1*). Although there is another TSS (TSS2), the activities of constructs P8 to P11 were relatively low in HepG2 cells (*Figure 5C*), suggesting that this alternative TSS may be minimally utilized. However, the activity of the P4 construct was relatively high (*Figure 5C*), suggesting that there are enhancers in P4. Notably, the activities of constructs P1, P7 and P4 were obviously increased by poly(I:C) stimulation, indicating that activating elements are mainly located within P1 (BS1) and P4 (BS4).

To assess the effect of STAT3 on P1, P7 and P4, we performed both knockdown and overexpression assays. As expected, knockdown of STAT3 in HepG2 cells upregulated the activities of all three constructs (*Figure 5D*, *Figure 5—source data 2*), whereas overexpression of STAT3 in 293FT cells (in which the endogenous p-STAT3 level is relatively low) inhibited their activities (*Figure 5E*, *Figure 5—source data 3*). To investigate whether STAT3 acts directly on the predicted binding sites, we generated mutant constructs (P1-M and P4-M, *Figure 5F*) and then performed similar assays. In HepG2 cells, mutation of the predicted STAT3-binding site within P1 resulted in an obvious decrease in the reporter activity and abolished the response of the P1 promoter to poly(I:C) stimulation (*Figure 5G*, *Figure 5—source data 4*), suggesting that this site is also essential for the transcriptional activation of P1 in HepG2 cells. Although the mutant P4 promoter displayed a reduced activity when compared to the wildtype promoter in the normal condition, it displayed an elevated activation when cells were stimulated with poly(I:C). In line with the hypothesis that STAT3 binds to the predicted sites on P1 and P4, the mutations reduced the activating effect of STAT3 knockdown on both promoters (*Figure 5G*). Intriguingly, probably because the context in 293FT cells differed in

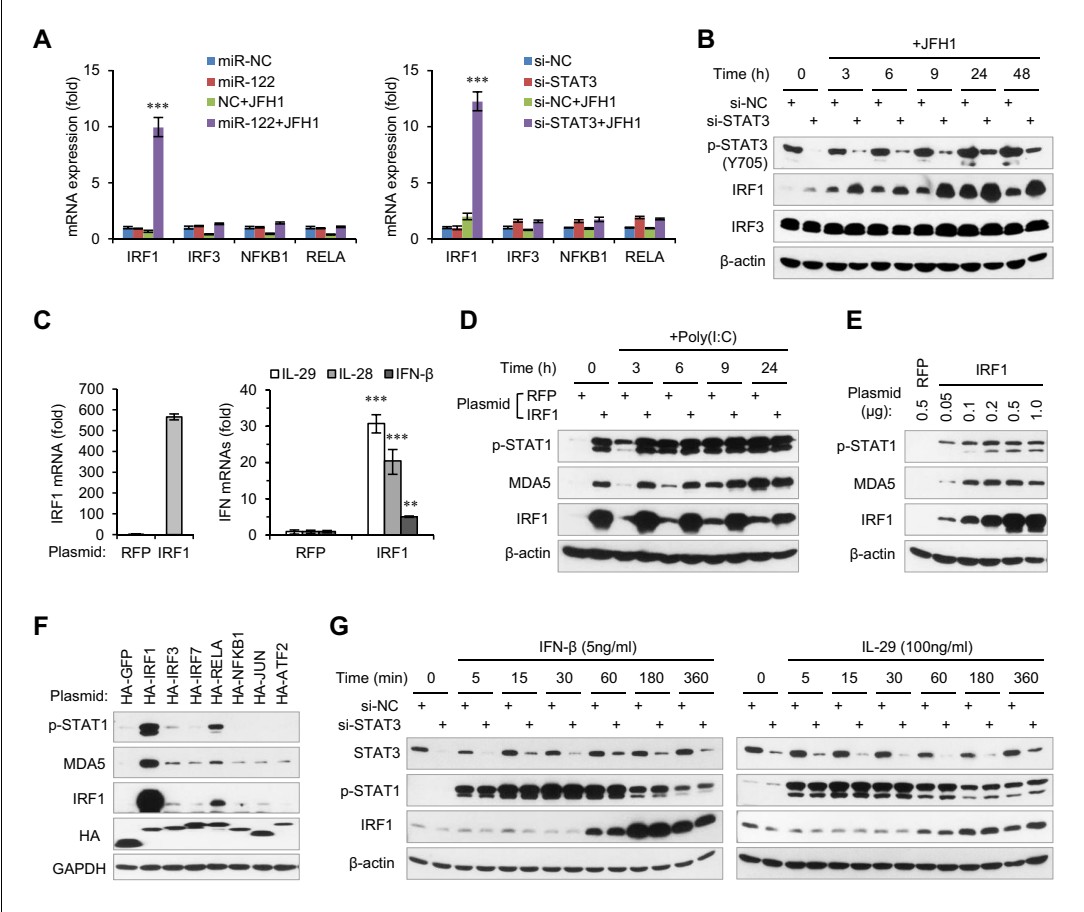

**Figure 4.** STAT3 inhibits the transcriptional activation of IRF1. (**A**) qRT-PCR analysis of IRF1, IRF3, NFKB1 and RELA in HepG2 cells first treated with mimics or siRNAs, and then transfected with or without JFH1 RNA for 24 hr. (**B**) Analysis of IRF1 and IRF3 protein expression in HepG2 cells treated with siRNAs and then JFH1 RNA. (**C**) qRT-PCR analysis of IRF1 and IFNs in HepG2 cells transfected with vectors expressing IRF1 or RFP (after 2 days). (**D**) Analysis of p-STAT1 and MDA5 in HepG2 cells transfected with IRF1 or RFP plasmids for 2 days, and then treated with poly(I:C) for 3–24 hr. (**E**) Analysis of p-STAT1 and MDA5 in HepG2 cells transfected with the indicated doses of IRF1 plasmids (0.05–1 µg/well in a 24-well-plate) for 2 days. (**F**) Analysis of IRF1, p-STAT1 and MDA5 in HepG2 cells transfected with plasmids expressing 7 HA-tagged transcription factors (after 2 days). HA-GFP was used as a negative control. (**G**) Analysis of IRF1 and p-STAT1 in HepG2 cells first transfected with STAT3 siRNA for 2 days, and then treated with IFN-β or IL-29 for 5–360 min. qRT-PCR data are from one experiment that was representative of three experiments (mean ± SEM of technical triplicates). *p<0.05, **p<0.01 and ***p<0.001.

DOI: https://doi.org/10.7554/eLife.41159.030

The following source data is available for figure 4:

**Source data 1.** qRT-PCR analysis of transcription factors in HepG2 cells.
DOI: https://doi.org/10.7554/eLife.41159.031
**Source data 2.** qRT-PCR analysis of IRF1 and IFN in HepG2 cells transfected with IRF1 plasmid.
DOI: https://doi.org/10.7554/eLife.41159.032

striking ways from that in HepG2 cells, mutant promoters displayed stronger activities than wildtype promoters in 293FT cells (*Figure 5H*, *Figure 5—source data 5*), evidently indicating a repressing role of STAT3 at these two sites.

In both cell lines, however, the repressing effect of STAT3 on P1 or P4 was not completely abolished by the mutations (*Figure 5G,H*), suggesting that STAT3 might still bind to mutant P1 and P4, through other sites or mechanisms. To validate this possibility, we transfected wildtype or mutant reporter constructs into HepG2 cells, and then performed STAT3 ChIP assays with PCR primers that can specifically detect the IRF1 promoters on the reporter constructs (*Figure 5—figure supplement 1B*). As expected, the mutations (P1-M and P4-M) obviously reduced the binding of STAT3 to both P1 and P4 promoters (*Figure 5I*), but STAT3 still binds to the mutant promoters, especially that of

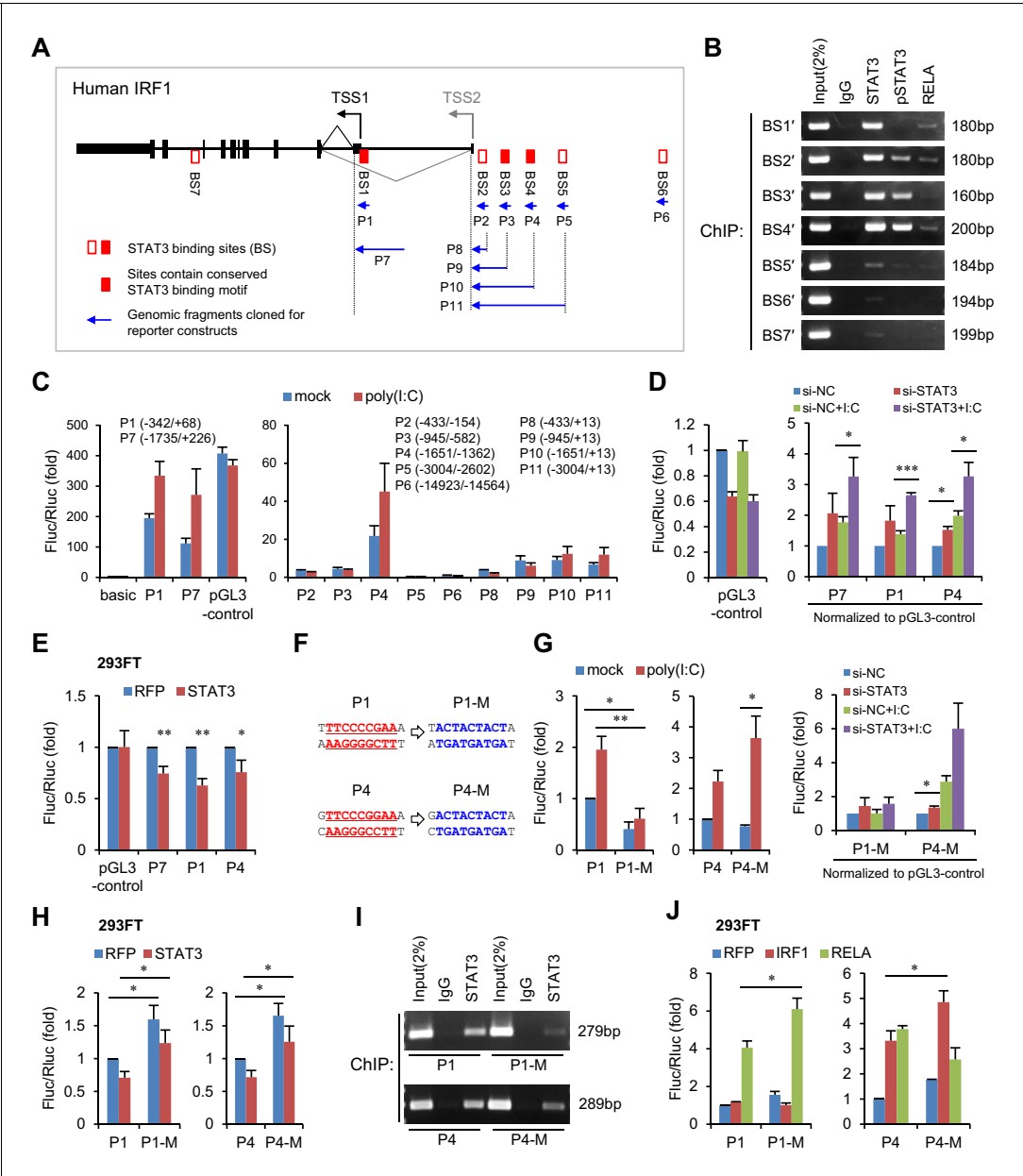

**Figure 5.** STAT3 directly binds to the promoter and enhancers of the IRF1 gene. (A) Schematic representation of STAT3-binding clusters (BS1–BS7) on the human IRF1 gene. The DNA fragments selected for reporter constructs (P1–P11) are also shown. (B) ChIP-PCR assays show the binding of STAT3, p-STAT3 and RELA on the selected gene fragments of IRF1 in HepG2 cells. BS1'–BS7' are short fragments (160–200 bp) corresponding to the BS1–BS7 clusters (290–410 bp), respectively. (C) Luciferase activity of different IRF1 promoter or enhancer constructs (P1–P11) in HepG2 cells treated with or without poly(I:C) (for 24 hr). The relative luciferase activities are the ratio of Firefly/Renilla luciferase normalized to the pGL3-basic vector (basic) without poly(I:C) stimulation. (D) Luciferase activity of P1, P7 and P4 constructs in HepG2 cells co-transfected with 20 nM STAT3 or control siRNAs for 2 days, and then treated with or without poly(I:C) for 24 hr. The fold changes of P1, P4 and P7 were normalized to the activity of the pGL3-control vector. (E) Luciferase activity of P1, P7 and P4 constructs in 293FT cells co-transfected with STAT3 or RFP plasmids (after 2 days). (F) The sequences of the STAT3 binding sites in wildtype (P1, P4) and mutant (P1-M, P4-M) constructs. (G, H) Luciferase activity of reporter constructs in HepG2 cells (G) or in 293FT cells (H) treated as in panels C, D and E, respectively. (I) ChIP-PCR assays show the binding of STAT3 to the wildtype or mutant reporter constructs in HepG2 cells. The forward primers that were used are the same as those used in panel B (BS1' and BS4'), but GLprimer2 was used as the reverse primer. (J) Luciferase activity of reporter constructs in 293FT cells co-transfected with plasmids expressing the indicated proteins (after 2 days). Luciferase data are from two (C) or three (D, E, G, H and J) experiments (mean + SD). *p<0.05, **p<0.01 and ***p<0.001.
DOI: https://doi.org/10.7554/eLife.41159.033

The following source data and figure supplement are available for figure 5:

**Source data 1.** Luciferase activity of different IRF1 promoter or enhancer constructs in HepG2 cells.

*Figure 5 continued on next page*

*Figure 5 continued*

DOI: https://doi.org/10.7554/eLife.41159.035

**Source data 2.** Luciferase activity of constructs in HepG2 cells co-transfected with STAT3 or control siRNAs.

DOI: https://doi.org/10.7554/eLife.41159.036

**Source data 3.** Luciferase activity of constructs in 293FT cells co-transfected with STAT3 or RFP plasmids.

DOI: https://doi.org/10.7554/eLife.41159.037

**Source data 4.** Luciferase activity of mutant constructs in HepG2 cells.

DOI: https://doi.org/10.7554/eLife.41159.038

**Source data 5.** Luciferase activity of mutant constructs in 293FT cells.

DOI: https://doi.org/10.7554/eLife.41159.039

**Source data 6.** ChIP-qPCR assays of BS1′ and BS4′ fragments bound by STAT3.

DOI: https://doi.org/10.7554/eLife.41159.040

**Source data 7.** Luciferase activity of constructs in 293FT cells co-transfected with the indicated plasmids.

DOI: https://doi.org/10.7554/eLife.41159.041

**Figure supplement 1.** Predicted STAT3, NFκB and IRF1 binding sites on the human IRF1 gene.

DOI: https://doi.org/10.7554/eLife.41159.034

P4. Previous studies have suggested that STAT3 could regulate gene transcription by physically interacting with other transcription factors, especially with NF-κB subunits (*Yang et al., 2007*; *Yu et al., 2002*). Given that NF-κB p65 (RELA) could bind to P1 and P4 (BS1′ and BS4′, *Figure 5B*), and that there are consensus or non-canonical NF-κB binding sites in P1 and P4 (*Figure 5—figure supplement 1C,D*), we further performed ChIP assays on BS1′ and BS4′ in HepG2 cells transfected with siRNAs that knockdown RELA or STAT3. Interestingly, RELA knockdown significantly reduced the binding of STAT3 to BS1′, even more effectively than knockdown of STAT3 itself (*Figure 5—figure supplement 1E*, *Figure 5—source data 6*), suggesting that STAT3 could also bind to P1 in a RELA-dependent manner. By contrast, RELA knockdown only slightly reduced the binding of STAT3 to BS4′, suggesting that STAT3 might interact with other transcriptional factors to bind P4.

To further understand how STAT3 inhibits IRF1 transcriptional activation, we co-transfected the P1 and P4 constructs with vectors overexpressing RELA or IRF1 (IRF1 may also regulate its own transcription through a consensus IRF-binding site in P4, *Figure 5—figure supplement 1D*) into 293FT cells. As expected, mutating the STAT3-binding sites significantly increased the activating effects of RELA and IRF1 on P1 and P4, respectively (*Figure 5J*, *Figure 5—source data 7*). Interestingly, STAT3 binding site mutations in P4-M blocked the activating effects of RELA on P4 (*Figure 5J*), probably because the mutation also influenced RELA binding to non-canonical NF-κB binding sites (*Figure 5—figure supplement 1D*). Taken together, these results demonstrate that STAT3 represses IRF1 activation by binding to the promoter and enhancers of the IRF1 gene.

## A group of genes mediated miR-122 regulation of STAT3 and IFNs

To search for the genes that mediate miR-122 regulation of p-STAT3, we first performed microarrays to identify the genes that are repressed by miR-122, and then screened these genes for STAT3 activators (*Figure 6A*). To obtain the genes that are truly regulated by miR-122, we performed both transient and stable miR-122 overexpression. For stable miR-122 overexpression, we generated a miR-122-Tet-On cell line, in which miR-122 expression can be strikingly induced by doxycycline treatment (*Figure 6—figure supplement 1A–C*, *Figure 6—source data 1*). By analysis of the intersection between genes that are downregulated by miR-122 (1141 genes were downregulated more than 20% by both transient and stable miR-122 overexpression) and the 330 potential STAT3 activators (*Supplementary file 1*), we obtained 25 candidates (*Supplementary file 2*). The qRT-PCR data confirmed that 20 genes were obviously downregulated by miR-122 (*Figure 6B*, *Figure 6—source data 2*), but the remaining five genes were barely detected or were unchanged (data not shown).

To determine whether these genes mediated the effect of miR-122 on p-STAT3 and the IFN response, we first performed knockdown experiments (*Figure 6—figure supplement 2*, *Figure 6—source data 3*). While knocking down most genes (except for FGFR1, DSTYK, ABL2 and OSMR) reduced the p-STAT3 level, the knockdown of ten genes (MERTK, EPO, FGFR1, JAK3, PKM2, IGF2, IL8, ABL2, MAK3K3 and FGF11) appeared to increase the p-STAT1 level (*Figure 6C*). Notably, some of these genes (MERTK, EPO, JAK3, PKM2 and IL8) significantly affected both p-STAT1 and

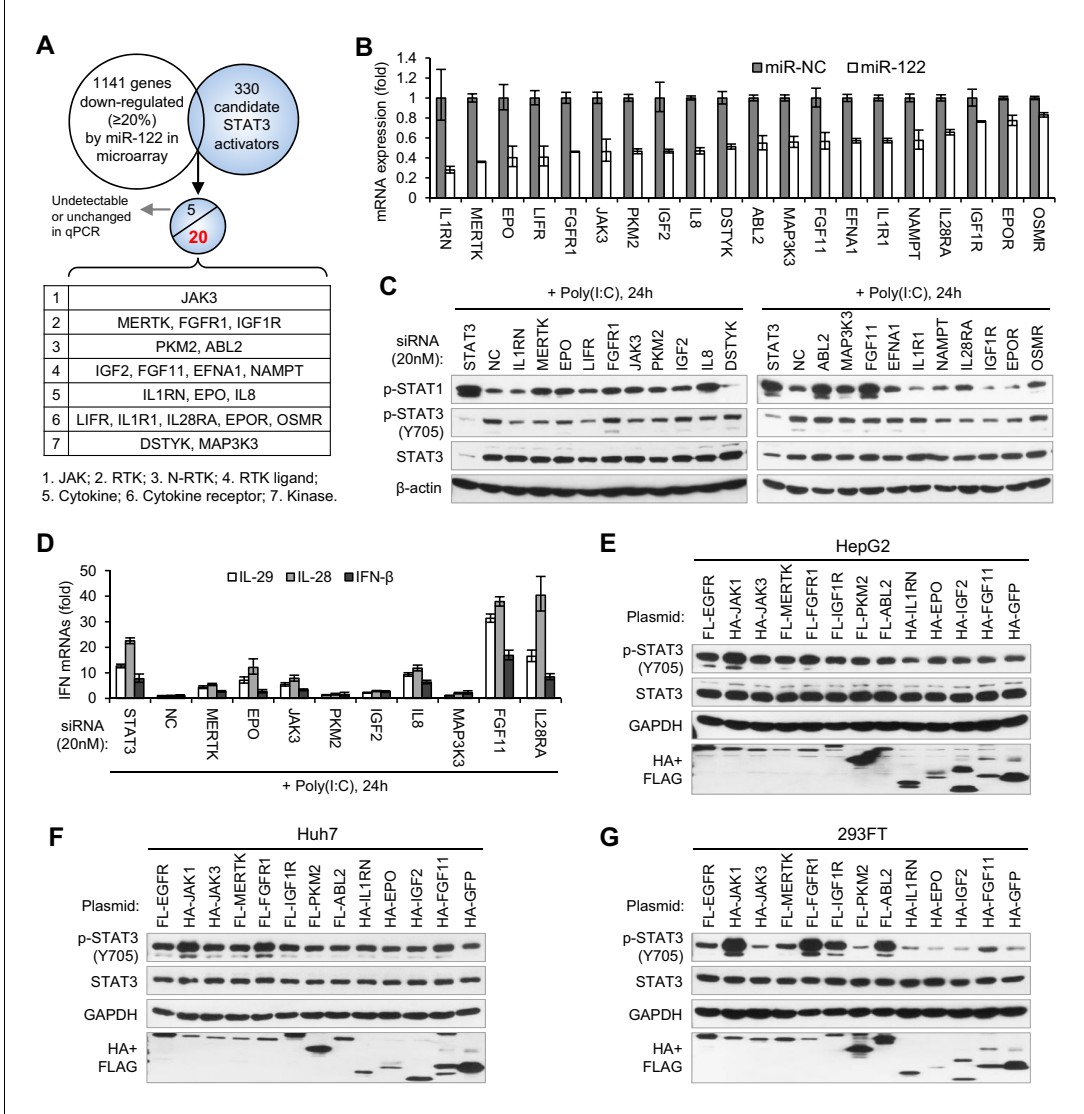

**Figure 6.** Identification of genes mediating miR-122 regulation of STAT3 and IFNs. (A) Strategy for identifying genes that mediated miR-122 regulation of STAT3. (B) qRT-PCR analysis of the 20 genes in HepG2 cells transfected with miR-122 or NC mimics (50 nM). The genes are ranked by the repression ratio. (C) Analysis of p-STAT1, p-STAT3 and total STAT3 protein in HepG2 cells first treated with the indicated siRNAs (20 nM) for 2 days, and then transfected with poly(I:C) for 24 hr. (D) qRT-PCR analysis of IFNs in HepG2 cells treated with siRNAs and poly(I:C), as in panel C. (E–G) Analysis of total and phosphorylated STAT3 in HepG2 (E), Huh7 (F) and 293FT (G) cells transfected with plasmids expressing HA- or Flag (FL)-tagged proteins. HA-JAK1 and FL-EGFR plasmids were employed as positive controls. HA-GFP was used as a negative control. qRT-PCR data are from one experiment that was representative of three experiments (mean ± SEM of technical triplicates).

DOI: https://doi.org/10.7554/eLife.41159.042

The following source data and figure supplements are available for figure 6:

**Source data 1.** qRT-PCR analysis of miR-122 levels in HepG2, Huh7, and miR-122-Tet-On cells.
DOI: https://doi.org/10.7554/eLife.41159.046
**Source data 2.** RT-PCR analysis of the 20 genes in HepG2 cells transfected with miR-122 or NC mimics.
DOI: https://doi.org/10.7554/eLife.41159.047
**Source data 3.** qRT-PCR analysis of the effectiveness of siRNAs.
DOI: https://doi.org/10.7554/eLife.41159.048
**Source data 4.** qRT-PCR analysis of IFNs in HepG2 cells treated with siRNAs and poly(I:C).
DOI: https://doi.org/10.7554/eLife.41159.049
**Figure supplement 1.** The miR-122-Tet-On cell line.
DOI: https://doi.org/10.7554/eLife.41159.043
**Figure supplement 2.** qRT-PCR analysis of the effectiveness of siRNAs in reducing the expression of the corresponding genes.
*Figure 6 continued on next page*

*Figure 6 continued*

DOI: https://doi.org/10.7554/eLife.41159.044

**Figure supplement 3.** The effects of eight tyrosine kinases on the phosphorylation of STAT3 and STAT1.

DOI: https://doi.org/10.7554/eLife.41159.045

p-STAT3. The qRT-PCR data confirmed that p-STAT1 upregulation was accompanied by an increase in IFN expression (*Figure 6D*, *Figure 6—source data 4*).

Next, we performed overexpression experiments on ten genes, which were selected because their knockdown either significantly increased p-STAT1 levels (FGFR1, IGF2, ABL2 and FGF11), significantly reduced p-STAT3 levels (IL1RN and IGF1R), or both (MERTK, EPO, JAK3 and PKM2). JAK1 and EGFR, two well-characterized STAT3 activators, were employed as positive controls. Most probably because STAT3 was constitutively phosphorylated in HepG2 cells, the overexpression of most of these genes did not further increase the p-STAT3 level in HepG2 cells (*Figure 6E*). In Huh7 or 293FT cells, however, the overexpression of most of the genes increased p-STAT3 levels (*Figure 6F, G*). In particular, FGFR1, IGF1R, MERTK, JAK3 and FGF11 clearly upregulated p-STAT3 levels (*Figure 6F,G*), even though the effect was slightly different in the two cell lines. Interestingly, the overexpression of RTKs, such as FGFR1, IGF1R and ABL2, also increased p-STAT1 (*Figure 6— figure supplement 3*), which may explain why STAT1 is phosphorylated in HepG2 cells without viral nucleic acid treatment and why miR-122 initially inhibited p-STAT1 (*Figure 2D*). Taken together, these results demonstrate that miR-122 may regulate STAT3 phosphorylation and the IFN response by repressing several STAT3 activators.

## miR-122 targets RTK signaling

Among the 20 genes that are downregulated by miR-122 (*Figure 6B*), IGF1R (*Bai et al., 2009*) and PKM2 (*Liu et al., 2014*) are known targets of miR-122. In addition, we have previously shown that MAP3K3 has a functional miR-122 binding site (*Xu et al., 2010*). Using Targetscan (*Agarwal et al., 2015*) or RegRNA (*Huang et al., 2006*), we found that 10 further genes from among the 20 downregulated genes possess potential miR-122-binding sites in their 3'UTR and two genes (MERTK and JAK3) have potential binding sites in their coding sequences (CDS) (*Figure 7—figure supplement 1*). By contrast, five genes, LIFR, IL8, NAPMT, IL28RA and EPOR, lack potential binding sites, suggesting that miR-122 repressed their expression indirectly. To investigate whether the predicted sites mediated the effect of miR-122 on these genes, we generated reporter constructs and performed luciferase assays. As positive controls, we also performed assays on IGF1R (two sites), PKM2 and MAP3K3. In addition to the four positive controls, seven sites from six genes (IL1RN, MERTK, EPO, FGFR1, DSTYK and FGF11) could be significantly repressed by miR-122 and mutating of their 'seed' sequence largely or completely abolished the response to miR-122 (*Figure 7A*, *Figure 7— source data 1*). Therefore, nine genes, including the three known miR-122 targets, were directly repressed by miR-122.

To identify the genes that play key roles in mediating the regulation of miR-122 on p-STAT3, we assessed the expression of all 20 genes in HepG2 and Huh7 cells and in normal human liver. Interestingly, the mRNA expression of six genes (IGF1R, PKM2, FGFR1, FGF11, IGF2 and MERTK) was significantly higher (>2 fold) in the HepG2 cells than in the normal liver (*Figure 7B*, *Figure 7—source data 2*), suggesting that these genes might account for the persistent STAT3 activation in HepG2 cells. To determine which gene was most influential, we again performed knockdown experiments with a higher dose of siRNAs (50 nM). Remarkably, while PKM2 and IGF2 knockdown moderately decreased p-STAT3 levels, MERTK and IGF1R knockdown markedly decreased the p-STAT3 level (*Figure 7C*), indicating that MERTK and IGF1R might be the key factors that mediate miR-122's regulation of p-STAT3. Although FGFR1 and FGF11 knockdown alone did not significantly reduce the p-STAT3 level, in view of the overexpression experiment results (*Figure 6F,G*), these genes might also be responsible for the constitutive phosphorylation of STAT3 in HepG2 cells. Consistent with the decreased mRNA expression, miR-122 overexpression sharply inhibited the protein expression of three RTKs: MERTK, FGFR1 and IGF1R (*Figure 7D*). Therefore, these results suggest that miR-122 suppresses STAT3 phosphorylation mainly by targeting RTK signaling.

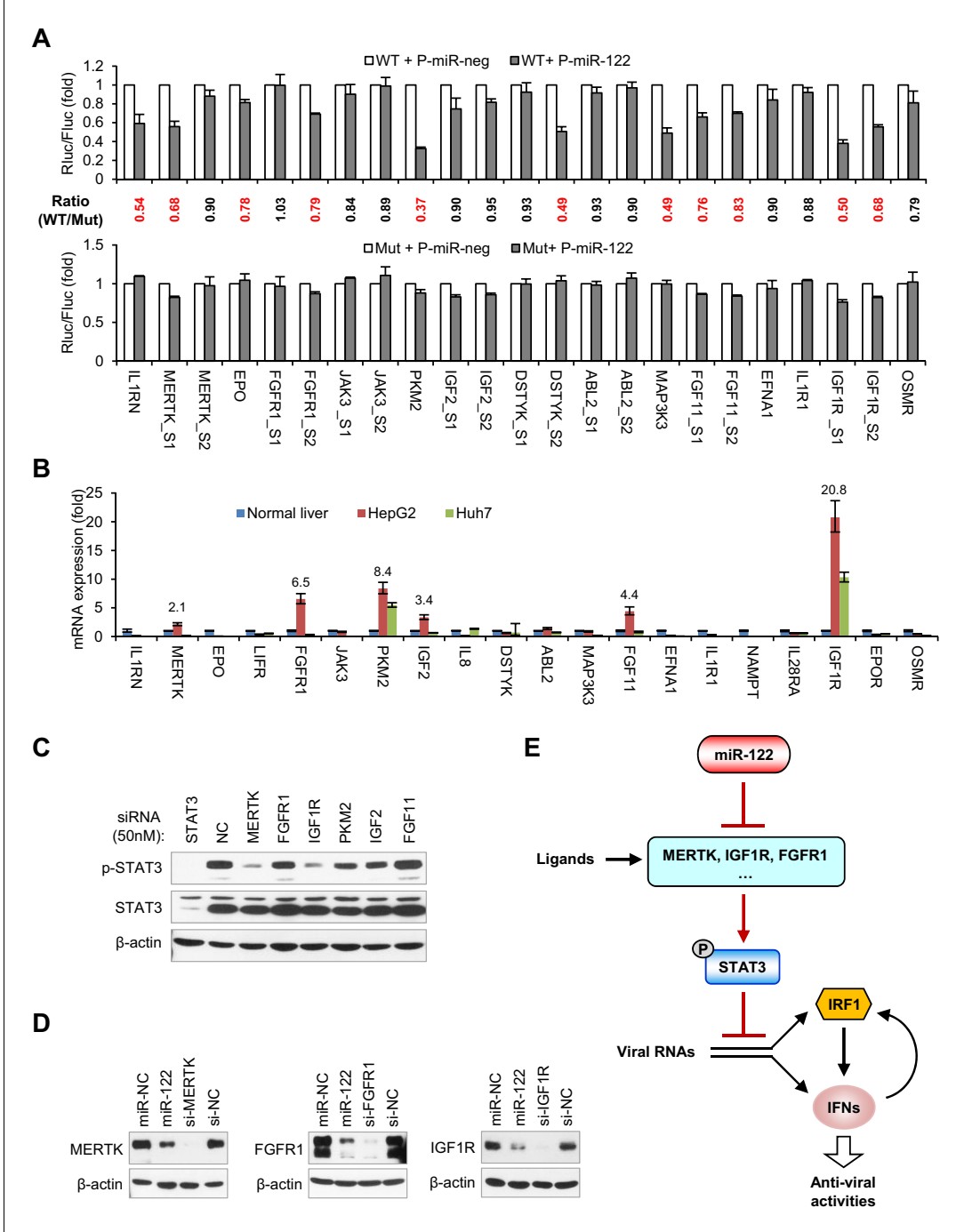

**Figure 7.** miR-122 targets RTK signaling to regulate STAT3 phosphorylation. (**A**) Luciferase activity of wildtype (WT) and mutant (Mut) reporter constructs in 293FT cells co-transfected with pcDNA6.2-miR-122 (P-miR-122) or pcDNA6.2-miR-neg (P-miR-neg) plasmids. The relative luciferase activities are the ratio of Renilla/Firefly luciferase normalized to that in P-miR-neg groups. The repression ratios (WT/Mut) of each site are shown and sites that significantly (p<0.05) repressed by miR-122 are shown in red. (**B**) qRT-PCR analysis of the 20 genes in normal human liver, HepG2 and Huh7. The expression of each gene was normalized to its level in normal liver. (**C**) Analysis of total and phosphorylated STAT3 in HepG2 cells transfected with indicated siRNAs (50 nM) for 2 days. (**D**) Analysis of MERTK, FGFR1 and IGF1R expression in HepG2 cells treated with miR-122 mimics or specific siRNAs (for 2 days). (**E**) Illustration of the mechanism by which miR-122 regulates STAT3 phosphorylation and the IFN response. Phosphorylated STAT3 can inhibit IRF1 transcriptional activation and thus represses the induction of IFNs upon viral infection. In normal hepatocytes, miR-122 strongly limits the phosphorylation of STAT3 by targeting three RTKs and other STAT3 activators, enabling a robust IFN response upon infection. qRT-PCR data are from one experiment that was representative of three experiments (mean ± SEM of technical triplicates). Luciferase data are from three experiments (mean +SD).

*Figure 7 continued on next page*

*Figure 7 continued*

DOI: https://doi.org/10.7554/eLife.41159.050

The following source data and figure supplements are available for figure 7:

**Source data 1.** Luciferase activity of reporter constructs in 293FT cells co-transfected with miR-122 or negative control plasmids.
DOI: https://doi.org/10.7554/eLife.41159.053
**Source data 2.** qRT-PCR analysis of the 20 genes in normal human liver, HepG2 and Huh7.
DOI: https://doi.org/10.7554/eLife.41159.054
**Source data 3.** qRT-PCR analysis of the effects of STAT3 knockdown on the expression of 20 genes in HepG2 cells.
DOI: https://doi.org/10.7554/eLife.41159.055
**Figure supplement 1.** miR-122-binding sites in the 3′UTR or the CDS of candidate genes, predicted by TargetScan or RegRNA.
DOI: https://doi.org/10.7554/eLife.41159.051
**Figure supplement 2.** qRT-PCR analysis of the effects of STAT3 knockdown on the expression of 20 genes in HepG2 cells transfected with 20 nM siRNAs (NC or STAT3) for 2 days.
DOI: https://doi.org/10.7554/eLife.41159.052

Given that STAT3 is a transcription factor that has the potential to regulate a variety of genes, we also investigated whether STAT3 regulates the 20 genes that we identified. Surprisingly, knockdown of STAT3 reduced the expression of half of these genes to different extents (*Figure 7—figure supplement 2*, *Figure 7—source data 3*), suggesting that STAT3 also regulates them in a positive feedback manner. Interestingly, within the genes that were downregulated by STAT3 knockdown, several genes lack functional miR-122-binding sites (JAK3, IL1R1, IGF2, EFNA1 and LIFR), suggesting that miR-122 indirectly affects their expression through the repression of p-STAT3. By contrast, several genes that have functional miR-122-binding sites, such as FGFR1, EPO, IL1RN, FGF11 and MERTK, were also downregulated by STAT3 knockdown, indicating that miR-122 represses their expression both directly and indirectly. In combination, these results suggest that miR-122 regulates a complex gene network sustaining STAT3 phosphorylation by directly targeting several key nodes within the network.

## Discussion

This study demonstrates that miR-122 promotes IFN-based innate immunity by regulating genes that contribute to the STAT3 phosphorylation level and thereby removing the negative regulation of STAT3 on IFN signaling. Our findings thus reveal a critical role for miR-122 in hepatocyte innate immunity, as illustrated in *Figure 7E*. According to this model, miR-122 is responsible for restricting STAT3 phosphorylation to a low level in normal hepatocytes, which enables a robust innate immune defense upon infection.

The role of miR-122 in hepatocyte innate immunity has remained undiscovered for a long time, probably because miR-122 appears to play completely different roles in HCV and HBV infections; miR-122 has an essential role in HCV replication especially, opposing a potential role of miR-122 in antiviral defense. Nevertheless, previous studies have provided some evidence supporting the role of miR-122 in IFN signaling. In chimpanzees with chronic HCV infection, antisense-mediated inhibition of miR-122 resulted in a rapid decline of the endogenous IFN pathway in the liver (*Lanford et al., 2010*). Although the decline of the IFN response may be attributed mainly to the downregulation of HCV RNA, this finding is inconsistent with our conclusion that miR-122 supports robust innate immunity. Further evidence to support such role are data showing that the expression of interferon-regulated genes was reduced to the normal level even when therapy did not completely eradicate detectable viral RNA (*Lanford et al., 2010*). This finding suggests that the decline in the IFN response preceded the decline of viral RNA. In addition, liver biopsy studies showed that decreased miR-122 levels in individuals with hepatitis C are associated with a poor response to IFN therapy (*Sarasin-Filipowicz et al., 2009*; *Urban et al., 2010*), suggesting that abundant miR-122 in hepatocytes promotes cellular responses to IFN treatment. As phosphorylated STAT3 is known to inhibit type I IFN-induced gene expression (*Ho and Ivashkiv, 2006*; *Wang et al., 2011*), our findings suggest that the decreased miR-122 in HCV-infected cells could lead to the upregulation of p-STAT3, thereby resulting in IFN resistance.

Although miR-122 has been recognized as a bona-fide tumor suppressor, the mechanisms that underlie this suppression remain to be fully understood. As STAT3 is a well-known oncoprotein that is persistently activated in many cancers, including HCC (*He and Karin, 2011*; *Yu et al., 2009*), our findings suggest that repressing STAT3 phosphorylation may be another mechanism that is essential for miR-122 to suppress tumorigenesis. Moreover, as miR-122 is frequently downregulated or lost in HCC, our findings also suggest that the loss of miR-122 may be a critical factor causing constitutive STAT3 activation in HCC cells.

Although a report suggested that miR-122 may directly target STAT3 (*Xiong et al., 2015*), this potential direct regulation was not observed in our study. Instead, our results suggest that miR-122 represses STAT3 phosphorylation by targeting STAT3 activators, such as MERTK, FGFR1 and IGF1R. Supportive of such mechanism is the evidence that miR-122 deletion in mice resulted in an upregulation of FGFR1 and IGF1R, but not of STAT3 (*Tsai et al., 2012*). Given that hepatocytes are continually exposed to various growth factors in the blood, which are produced either by the liver (such as IGF-I (*Sjögren et al., 1999*)) or by other organs, tightly controlling the expression of their receptors is crucial for hepatocytes to avoid excessive activation of the related pathways. Therefore, our findings further suggest that the suppression of these RTKs may be a critical mechanism that allows miR-122 to regulate liver homeostasis.

Although our screening method was able to reveal the STAT3 regulators that are evidently repressed by miR-122 (their mRNA expression was reduced), it should be noted that the approach might miss some important STAT3 regulators that are only downregulated at the protein level, as miRNAs sometimes only affect the translation of their targets. Through analysis of the candidate miR-122 targets identified in published CLIP-seq data using starBase (*Li et al., 2014*), we found that there are still dozens of potential STAT3 activators that might be directly regulated by miR-122 (*Supplementary file 3*). Moreover, given that STAT3 could regulate many of its activators in a positive-feedback manner (*Figure 7—figure supplement 2*), the regulation of the STAT3 signaling pathway by miR-122 may still be far from fully understood.

When we investigated miR-122's roles in different liver cell lines (*Figure 3—figure supplement 2*), we found that miR-122 promoted the IFN response in Huh7 cells, but did not affect the expression of p-STAT3, suggesting that miR-122 might also regulate innate immunity through other mechanisms. However, comparing the effects of miR-122 overexpression and STAT3 knockdown on IFN responses in the HepG2, Huh7 and Hep3B cell lines confirmed that repressing STAT3 phosphorylation should be the key mechanism through which miR-122 regulates innate immunity. Given that these cell lines are derived from HCC patients with different backgrounds, further understanding of the differences observed in these cells would shed new light on the roles of both miR-122 and STAT3 in liver biology and disease.

STAT3 is known to block the innate and adaptive immune responses in diverse cell types (*Ho and Ivashkiv, 2006*; *Wang et al., 2004*; *Wang et al., 2011*), but the mechanism has remained elusive. Our study identified that STAT3 limits the IFN responses by repressing IRF1, a key transcriptional activator of IFNs. Moreover, since IRF1 is a special IFN-signaling effector, which upon expression directly translocates to the nucleus to enhance the expression of a subset of ISGs (*Schoggins et al., 2011*), our findings suggest that IRF1 may mediate a considerable portion of STAT3's effects on immune responses. Intriguingly, both phosphorylated and unphosphorylated STAT3 were able to bind to the IRF1 promoter and enhancers, and the binding of phosphorylated STAT3 increased progressively from proximal to distal binding sites (BS1′ to BS4′, *Figure 5B*), suggesting that STAT3 regulates IRF1 transcription through a complex mechanism that should be further investigated.

Previous studies have revealed that STAT3 can be activated by different mechanisms during HCV infection (*Gong et al., 2001*; *Yoshida et al., 2002*). Large-scale screenings have also identified STAT3 as one of the main host factors required for HCV infection (*Randall et al., 2007*). However, the key contribution of STAT3 to the HCV life cycle remains undefined. Our results suggest that the upregulation of STAT3 activity may reduce the IFN response and could thereby facilitate HCV infection. Interestingly, HCV can sequester miR-122 through a 'sponge' effect and globally reduces miR-122 targets (*Luna et al., 2015*), suggesting that HCV-induced miR-122 sequestration may be another mechanism that results in STAT3 activation and immune tolerance.

An interesting result that we observed is that HCV RNAs are barely increased (only 1.5 fold) by miR-122 overexpression (*Figure 2A*), whereas there is a massive increase in viral protein production (core and NS3) (*Figure 2D*). These data suggest that miR-122 might primarily contribute to HCV

translation (as has also been suggested by other studies (*Henke et al., 2008*; *Jangra et al., 2010*; *Roberts et al., 2011*)), rather than the protection of HCV RNA. This hypothesis was further supported by the quantitative assessment of the effect of miR-122 on HCV translation, using a HCV construct (SGR-JFH1/Gluc) carrying a luciferase reporter (*Figure 2—figure supplement 1A*). Although miR-122 overexpression led to a more than 20-fold increase in HCV translation (as determined by GLuc activity), knockdown of XRN1 (a 5′ exonuclease that was reported to mediate HCV decay (*Li et al., 2013b*)) only resulted in a 2.7-fold increase in GLuc activity (*Figure 2—figure supplement 1B*). As HCV proteins are not only essential for replication of the virus but also required for blocking the host immunity (*Li et al., 2005b*; *Li et al., 2005c*), we further hypothesize that the primary intention of HCV in binding miR-122 might be to recruit factors that promote the rapid translation of viral RNA, a key event that is essential for long-term viral survival.

In conclusion, this work identifies a previously unknown function for miR-122 in antiviral immune defense, which broadens and deepens our understanding of the roles of miR-122 in liver biology and disease. In particular, we demonstrate that miR-122 plays a dual role in the HCV life cycle, which should be paid attention to in treating HCV. On the basis of our finding that STAT3 is a key negative regulator in the miR-122–RTKs/STAT3–IRF1–IFNs circuitry, STAT3 targeting may be a promising strategy for treating hepatitis viruses. Finally, as miR-122 is a representative of the tissue-specific and/or highly abundant miRNAs, our findings further suggest that a wide range of miRNAs may be involved in the innate immune system by targeting and connecting to different nodes of the immune gene network in a tissue-specific manner, and this should be further demonstrated in the future.

# Materials and methods

## Key resources table

| Reagent type (species) or resource | Designation | Source or reference | Identifiers | Additional information |
|---|---|---|---|---|
| Cell line (*Homo sapiens*) | HepG2 | http://www.cellbank.org.cn/ | Cat. #:TCHu 72, RRID:CVCL_0027 | |
| Cell line (*H. sapiens*) | Huh7 | | | Dr. Shi-Mei Zhuang (Sun Yat-sen University, China) |
| Cell line (*H. sapiens*) | Huh7.5.1 | PMID: 15939869 | RRID: CVCL_E049 | Dr. Jin Zhong (Institute Pasteur of Shanghai, China) |
| Cell line (*H. sapiens*) | 293FT | Invitrogen | Cat. #: R700-07, RRID: CVCL_6911 | |
| Cell line (*H. sapiens*) | Hep3B | | | Dr. Shu-Juan Xie (Sun Yat-sen University, China) |
| Cell line (*Mus musculus*) | BNL CL.2 | http://www.cellbank.org.cn/ | RRID: CVCL_4383 | ATCC TIB-73, https://www.atcc.org/ |
| Cell line (*H. sapiens*) | miR-122-Tet-On HepG2 | This paper | | *Figure 6—figure supplement 1*; cell lines and tissue sample in the 'Materials and methods' |
| Antibody | Rabbit anti-Phospho-STAT1 (Tyr701), (D4A7) Rabbit mAb | Cell signaling | Cat. #: 7649, RRID:AB_10950970 | WB (1:3000) |

*Continued on next page*

Continued

| Reagent type (species) or resource | Designation | Source or reference | Identifiers | Additional information |
|---|---|---|---|---|
| Antibody | Rabbit anti-Phospho-STAT3 (Tyr705), (D3A7) Rabbit mAb | Cell Signaling | Cat. #: 9145, RRID:AB_2491009 | WB (1:3000) |
| Antibody | Mouse anti-Phospho-STAT3 (Tyr705), (B-7) mouse mAb | Santa Cruz Biotechnology | Cat. #: sc-8059, RRID:AB_628292 | IP (2 ug/500 ul) |
| Antibody | Rabbit anti-STAT3, polyclonal | Santa Cruz Biotechnology | Cat. #: sc-482, RRID:AB_632440 | WB (1:10000), IP (2 ug/500 ul) |
| Antibody | Rabbit anti-IRF1, (D5E4) Rabbit mAb | Cell signaling | Cat. #: 8478, RRID:AB_10949108 | WB (1:2000) |
| Antibody | Rabbit anti-IRF3, (D6I4C) Rabbit mAb | Cell signaling | Cat. #: 11904, RRID:AB_2722521 | WB (1:2000) |
| Antibody | Rabbit anti-RELA, polyclonal | Santa Cruz Biotechnology | Cat. #: sc-372, RRID:AB_632037 | IP (2 ug/500 ul) |
| Antibody | Rabbit anti-MDA5, (D74E4) Rabbit mAb | Cell signaling | Cat. #: 5321, RRID:AB_10694490 | WB (1:3000) |
| Antibody | Mouse anti-HCV NS3, [H23] mouse mAb | Abcam | Cat. #: ab13830, RRID:AB_300673 | WB (1:3000, in 1% milk/TBST) |
| Antibody | Mouse anti-HCV core, (C7-50) mouse mAb | Thermo Fisher | Cat. #: MA1-080, RRID:AB_325417 | WB (1:3000) |
| Antibody | Mouse anti-MERTK, (B-1) mouse mAb | Santa Cruz Biotechnology | Cat. #: sc-365499, RRID:AB_10843860 | WB (1:2000) |
| Antibody | Rabbit anti-FGFR1, (D8E4) Rabbit mAb | Cell signaling | Cat. #: 9740, RRID:AB_11178519 | WB (1:2000) |
| Antibody | Rabbit anti-IGF1Rβ, (111A9) Rabbit mAb | Cell signaling | Cat. #: 3018, RRID:AB_10848600 | WB (1:2000) |
| Recombinant DNA reagent | pJFH1 | PMID: 15951748 | | Dr. Takaji Wakita (National Institute of Infectious Diseases) |
| Recombinant DNA reagent | pSGR-JFH1 | PMID: 15951748 | | Dr. Takaji Wakita (National Institute of Infectious Diseases) |
| Recombinant NA reagent | pJFH1-M (S1-S2-p6m) | This paper | | Viral constructs and viral RNA preparation in the 'Materials and methods' |
| Recombinant DNA reagent | pSGR-JFH1/Gluc | This paper | | Viral constructs and viral RNA preparation in the 'Materials and methods' |

*Continued*

| Reagent type (species) or resource | Designation | Source or reference | Identifiers | Additional information |
|---|---|---|---|---|
| Recombinant DNA reagent | pGL3-IRF1-P1 | This paper | | Constructs in the 'Materials and methods' |
| Recombinant DNA reagent | pGL3-IRF1-P1-M | This paper | | Constructs in the 'Materials and methods' |
| Recombinant DNA reagent | pGL3-IRF1-P7 | This paper | | Constructs in the 'Materials and methods' |
| Recombinant DNA reagent | pGL3-IRF1-P4 | This paper | | Constructs in the 'Materials and methods' |
| Recombinant DNA reagent | pGL3-IRF1-P4-M | This paper | | Constructs in the 'Materials and methods' |

## Mimics, miRNA inhibitors and siRNAs

miR-122 (4464066) and control (4464058) mimics, as well as control (4464076) and miR-122 (4464084) inhibitors, were obtained from Ambion. Negative control siRNA were obtained from Dharmacon. Gene-specific siRNAs were obtained from RiboBio with three independent siRNAs for each gene. STAT3 siRNAs were: si-1 (GCCUCUCUGCAGAAUUCAA), si-2 (AGUCAGGUUGCUGGUCAAA) and si-3 (CCGUGGAACCAUACACAAA).

## Antibodies

Phospho-STAT1 (Tyr701, CST, 7649), Phospho-STAT3 (Tyr705, CST, 9145, for western blot), Phospho-STAT3 (Tyr705, sc-8059, for immunoprecipitation), STAT3 (sc-482), IRF1 (CST, 8478), IRF3 (CST,11904), RELA (sc-372), MDA5 (CST, 5321), HCV NS3 (Abcam, ab13830), HCV Core (Thermo Fisher, MA1-080), HA (sigma, H9658), FLAG (WAKO, 014–22383), MERTK (sc-365499), FGFR1 (CST, 9740), IGF1Rβ (CST, 3018), β-actin (CST, 4970) and GAPDH (CST, 2118).

## Reagents and kits

Poly(I:C) LMW (tlrl-picw), 3p-hpRNA (tlrl-hprna), HSV-60 (tlrl-hsv60c) and ssRNA40 (tlrl-lrna40) were obtained from InvivoGen. S3I-201 (S1155) and cryptotanshinone (S2285) were obtained from Selleck. Recombinant cytokines (IL-29, 1598-IL-025; IFN-β, 8499-IF-010; IL-6, 206-IL-010) and ELISA kits (IL-29/IL-28B, DY1598B; IFN-β, DY814) were obtained from R&D systems.

## Cell lines and tissue sample

The cell lines used were HepG2, Huh7, Huh7.5.1 (*Zhong et al., 2005*), HepG2-2.15 (*Sells et al., 1987*), Hep3B, BNL CL.2, and 293FT. Cell identity has been authenticated by STR profiling and no mycoplasma contamination was detected during the study. The miR-122-Tet-On HepG2 cell line was generated using lentiviral vectors (*Figure 6—figure supplement 1*). Lentiviruses were generated in 293FT cells using ViraPower Lentiviral Packaging Mix (Invitrogen, K4975-00). HepG2 cells were infected with lentiviruses one after another, and selected with G418 (1000 µg/ml) and Blasticidin (10 µg/ml) until the cells were no longer dying. The normal human liver sample was obtained from the Liver Transplant Center, The Third Affiliated Hospital of Sun Yat-sen University.

Viral constructs and viral RNA preparation pJFH1 and pSGR-JFH1 vectors have been described previously (*Wakita et al., 2005*). pJFH1-M was generated by the introduction of two point mutations (A-T) in the miR-122-binding sites on pJFH1, as described previously (*Li et al., 2013b*). pSGR-JFH1/Gluc (*Figure 2—figure supplement 1A*) was generated based on pSGR-JFH1, by replacing the Neomycin resistance gene with the Nano-Glo Luciferase. HCV RNAs were prepared by in vitro transcription (Ambion, AM1333) as described previously (*Zhong et al., 2005*). HepG2-2.15 cells harboring a stable HBV replicon and the total RNAs from this cell line contain HBV RNAs. The HBV 1.3-mer WT replicon vector (*Wang et al., 2009*) was obtained from Addgene and used for

generation of HBV RNAs by transfection of this vector into HepG2 cells. The ε-region of HBV RNA (εRNA) was prepared as described previously (*Sato et al., 2015*).

## Constructs

STAT3 reporter vector was included in Cignal Finder Multi-Pathway Reporter Array (SABiosciences, CCA-901L). The IRF1 promoter or enhance constructs were generated based on pGL3-basic, and the information on the length and the position of each DNA fragment can be found in *Figure 5A and C*. Reporter vectors for miR-122-binding sites were generated based on psiCHECK-2, as we described previously (*Xu et al., 2010*). Briefly, 47-nt DNA fragments containing the putative binding site for miR-122 of each gene (5′-end 29–30 nt flanking sequences, 7–8 nt seed-matched sequences, 3′-end 10-nt flanking sequences) were cloned into Xho I/Not I-digested psiCHECK-2 vector (Promega) in the forward direction. Mutated constructs were generated by replacing the miR-122 seed-matched sequence (CACTCC) with (GTGAGG). The coding sequences of protein genes were amplified by RT-PCR and cloned into pcDNA3 (IRF1, STAT3), pRKW2-HA (HA-IRF3, HA-IRF7, HA-RELA, HA-NFKB1, HA-JUN and HA-ATF2), pRKW2-Flag (FL-EGFR, FL-MERTK, FL-FGFR1, FL-IGF1R, FL-ABL2 and FL-PKM2), or pcDNA3.1-HA (HA-IRF1, HA-JAK1, HA-JAK3, HA-IL1RN, HA-EPO, IGF2-HA and HA-FGF11). All of the oligonucleotides used are shown in *Supplementary file 4*.

## Analysis of STAT3-binding sites on IRF1

ChIP-seq binding sites for STAT3 were analyzed on the UCSC genome browser (GRCh37/hg19) using the 'ENCODE Regulation Super-track'. ChIP was performed using the SimpleChIP Kit (CST, 9003). HepG2 cells from four 10 cm dishes were used in ChIP. Cells were starved overnight and then treated with IL-6 (25 ng/ml) for 30 min before harvest. In total, 500 µl of cross-linked chromatin were obtained, and each 100 µl (diluted into 500 µl by adding 400 µl 1X ChIP buffer) was immuno-precipitated with 2 µg of the following antibodies: Normal Rabbit IgG (CST, 2729), Phospho-STAT3 (Tyr705, sc-8059), STAT3 (sc-482) and RELA (sc-372). After determination of the appropriate amplification cycle numbers by qPCR, the IPed chromatins were analyzed by standard PCR with AccuPrime PFX SuperMix (Invitrogen, 12344–040). For ChIP assays on the promoters in reporter vectors, HepG2 cells ($6 \times 10^6$) were transfected with 10 µg of plasmids (P1, P4, P1-M and P4-M) in 10 cm dishes for two days, and then treated as above. In PCR, a GLprimer2 that binds to the coding sequence of firefly luciferase was used as the reverse primer.

## Collection of candidate STAT3 activators

330 candidates (*Supplementary file 1*) were collected as follows: (1) genes that are involved in the JAK-STAT signaling pathway (https://www.rndsystems.com/cn/pathways/jak-stat-signaling-pathway); (2) genes that promote STAT3 phosphorylation, identified by Komurov and colleagues ('Other Known Regulator') (*Komurov et al., 2010*); (3) all other known RTKs (*Lemmon and Schlessinger, 2010*) and N-RTKs (GeneCards); (4) all other interleukins ('Other Cytokine') and their receptors ('Other Cytokine Receptor'), as well as related binding proteins ('Cytokine-receptor interaction'); and (5) potential RTKs, ligands, and RTK-related genes.

## Cell culture and transfection

HepG2, HepG2-2.15 and miR-122-Tet-On HepG2 cells were maintained in MEM. Huh7, Huh7.5.1 and 293FT were maintained in DMEM. Mimics and siRNAs were transfected using Lipofectamine™2000 (Invitrogen) at a final concentration of 20 nM (for siRNAs, three independent oligos to each gene were mixed equally and 20 nM is the concentration of all mixed oligos), unless otherwise indicated. HCV RNAs were transfected into HepG2 cells using the TransIT-mRNA Transfection Kit (Mirus) and into Huh7 and Huh7.5.1 cells using Lipofectamine 2000, at a final concentration of 1 µg/ml. Poly(I:C) were transfected with Lipofectamine 2000 at a final concentration of 2 µg/ml. Plasmids were transfected into HepG2 and Huh7 using ViaFect Transfection Reagent (Promega) and into 293FT using Lipofectamine 2000.

## Reporter assays

For reporter assays on signaling pathways, HepG2 cells were transfected in 48-well-plates, with 20 nM mimics, 0.1 µg firefly luciferase reporter plasmids, and 0.0025 µg control plasmids (Renilla

luciferase); 48 hr later, cells were further transfected with SGR-JFH1 RNA (0.25 µg) for the indicated treatment durations. For comparison of 11 different IRF1 promoter or enhancer constructs, HepG2 cells were transfected in 48-well-plates, with 0.2 µg Firefly luciferase reporter plasmids and 0.005 µg control Renilla plasmids; 48 hr later, cells were further transfected with poly(I:C) for 24 hr. For assessment of the effects of STAT3 knockdown on IRF1 constructs, HepG2 cells were transfected in 48-well-plates with 20 nM control or STAT3 siRNAs (using Lipofectamine 2000), 0.1 µg reporter plasmids, and 0.1 µg pRL-TK plasmids (using ViaFect Transfection Reagent). For assessment of the effects of STAT3 and other transcription factors on IRF1 constructs, 293FT cells were transfected in 48-well-plates with 0.1 µg reporter plasmids, 0.1 µg transcription factor expressing plasmids, and 0.0025 µg control Renilla plasmids. For miR-122 target identification, 293FT cells were transfected in 96-well-plates with 0.05 µg reporter plasmids, 0.05 µg miR-122 or miR-neg expressing vectors. Firefly and Renilla luciferase activities were measured by Dual-Luciferase Reporter Assay System (Promega).

## qRT-PCR

The expression of mRNAs, HCV RNA and miR-122 were quantified by SYBR Green-based real-time PCR. Reverse transcription (RT) reactions for mRNAs were done using GoScript RT System (Promega, A5000), with random primers and oligo dT. RT reactions for miR-122 were performed using PrimeScript RT reagent Kit (TaKaRa, 047A), with a stem-loop primer. RT reactions for HCV RNA were performed using SuperScript III Reverse Transcriptase, with a specific primer. qPCRs were performed with GoTaq qPCR Master Mix (Promega, A6002).

## Western blot

Protein samples were prepared using RIPA buffer (50 mM Tris-HCl, pH 8.0, 150 mM Sodium chloride, 1% NP-40, 0.5% sodium deoxycholate, 0.1% sodium dodecyl sulfate (SDS), 2 mM EDTA) with 1X Protease Inhibitor Cocktail (Roche, 04693132001) and 1x PhosSTOP phosphatase inhibitor (Roche, 04906837001). Proteins were separated by SDS-PAGE and transferred onto Protran nitrocellulose membranes (Amersham, 10600001). HRP-conjugated secondary antibody (CST, 7074 or 7076) and Luminata Crescendo Western HRP substrate (Millipore, WBLUR0500) were used to detect the immunoreactive signals.

## Microarray

HepG2 cells transfected with 50 nM mimics (control or miR-122) were harvested 72 hr post-transfection. miR-122-Tet-On HepG2 cells were cultured 72 hr with or without doxycycline (1000 ng/ml). RNAs were sent to microarray analysis with GeneChip Human Genome U133 Plus 2.0 Arrays (CapitalBio Corporation). Microarray data have been deposited in GEO under accession number GSE99663.

## Statistical analysis

The differences between groups were analyzed using two-sided Student's $t$-tests. $*p<0.05$, $**p<0.01$ and $***p<0.001$.

## Acknowledgements

We thank Takaji Wakita (National Institute of Infectious Diseases, Japan) for use of pJFH1 and pSGR-JFH1, Francis Chisari (Scripps Research Institute, US) for use of Huh7.5.1 cell line, and Jin Zhong (Institute Pasteur of Shanghai, China) for providing these materials. We thank Hong-bing Shu (Wuhan University, China) for providing the FLAG and HA-tagged RKW2 vectors. We thank Jun Cui (Sun Yat-sen University, China) for helpful discussions.

## Additional information

### Funding

| Funder | Grant reference number | Author |
|---|---|---|
| National Natural Science Foundation of China | 31200593 | Hui Xu |
| National Natural Science Foundation of China | 31230042 | Liang-Hu Qu |
| National Natural Science Foundation of China | 31471223 | Liang-Hu Qu |
| National Natural Science Foundation of China | 31671349 | Liang-Hu Qu |
| Natural Science Foundation of Guangdong Province | 2014A030313163 | Hui Xu |
| National Basic Research Program of China | 2011CB811300 | Liang-Hu Qu |

The funders had no role in study design, data collection and interpretation, or the decision to submit the work for publication.

### Author contributions

Hui Xu, Conceptualization, Resources, Data curation, Software, Formal analysis, Funding acquisition, Validation, Investigation, Visualization, Methodology, Writing—original draft, Writing—review and editing; Shi-Jun Xu, Conceptualization, Data curation, Software, Formal analysis, Validation, Investigation, Visualization, Methodology, Writing—original draft, Writing—review and editing; Shu-Juan Xie, Data curation, Investigation, Writing—review and editing; Yin Zhang, Data curation, Software, Investigation, Writing—review and editing; Jian-Hua Yang, Software, Funding acquisition, Writing—review and editing; Wei-Qi Zhang, Man-Ni Zheng, Investigation, Writing—review and editing; Hui Zhou, Supervision, Project administration, Writing—review and editing; Liang-Hu Qu, Conceptualization, Supervision, Funding acquisition, Project administration, Writing—review and editing

### Author ORCIDs

Hui Xu http://orcid.org/0000-0003-3424-1219
Shi-Jun Xu http://orcid.org/0000-0003-3309-9791
Jian-Hua Yang https://orcid.org/0000-0003-3863-2786
Liang-Hu Qu http://orcid.org/0000-0003-3657-2863

### Decision letter and Author response

Decision letter https://doi.org/10.7554/eLife.41159.064
Author response https://doi.org/10.7554/eLife.41159.065

## Additional files

### Supplementary files

• Supplementary file 1. The 330 candidate STAT3 regulators.
DOI: https://doi.org/10.7554/eLife.41159.056

• Supplementary file 2. The expression of 25 candidate STAT3 activators in microarray data.
DOI: https://doi.org/10.7554/eLife.41159.057

• Supplementary file 3. Candidate STAT3 activators that are predicted to be miR-122 targets in published CLIP-seq data. The candidate miR-122 targets and binding sites were predicted by starbase (http://starbase.sysu.edu.cn/). The targets shown are 47 genes from among the 330 candidate STAT3 regulators.
DOI: https://doi.org/10.7554/eLife.41159.058

• Supplementary file 4. Oligonucleotides.

DOI: https://doi.org/10.7554/eLife.41159.059
• Transparent reporting form
DOI: https://doi.org/10.7554/eLife.41159.060

## Data availability

Microarray data have been deposited in GEO under accession number GSE99663.

The following dataset was generated:

| Author(s) | Year | Dataset title | Dataset URL | Database and Identifier |
|---|---|---|---|---|
| Xu H, Xu S-J, Xie S-J, Zhang Y | 2017 | MicroRNA-122 promotes antiviral interferon response by inhibition of phosphorylated STAT3 | https://www.ncbi.nlm.nih.gov/geo/query/acc.cgi?&acc=GSE99663 | NCBI Gene Expression Omnibus, GSE99663 |

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
