## [Decision Letter]

Thank you for submitting your article "MicroRNA-122 supports robust innate immunity in hepatocytes by targeting the RTKs/STAT3 signaling pathway" for consideration by *eLife*. Your article has been reviewed by three peer reviewers, and the evaluation has been overseen by a Reviewing Editor and Jonathan Cooper as the Senior Editor. The following individuals involved in review of your submission have agreed to reveal their identity: Yutaka Enomoto (Reviewer #3).

The reviewers have discussed the reviews with one another and the Reviewing Editor has drafted this decision to help you prepare a revised submission.

Summary:

In this complex manuscript, Xu et al. report that miR-122, the most highly expressed miRNA in hepatocytes, increases the activation of the interferon response in response to RNA PAMPs by down regulating the phosphorylation of STAT3, reducing IRF1 function. The primary effect of miR-122 is to down regulate several receptor tyrosine kinases (RTKs), including MER, FGFR1 and IGF1R, that are responsible for phosphorylation of STAT3.

Essential revisions:

Our reviewers have a number of criticisms that need to be addressed.

1) A concern is that the identification of the mRNA targets for miR-122 has involved some cherry picking. The hit list is long, and only a few are examined further. Other important hits may have been missed, and this should be acknowledged. It would be better to directly identify mRNAs targeted by miR-122 by performing some form of RISC CLIP experiment, such as iCLIP or PAR-CLIP, in the presence and absence of miR-122 using a pan-Ago specific antibody. This will identify all seed target sites for miR-122 occupied in vivo. I would not insist on this last experiment but it would strengthen the paper.

2) The average cell expresses ~50,000 miRNAs or so and miRNA function is dependent on expression level, with miRNAs expressed at <100 copies per cell being essentially non-functional (Mullokandov et al., 2012). Moreover, only RISC associated miRNAs are functional. In Figure 1B, the authors claim that human primary hepatocytes express 81,858 times more miR-122 that HepG2 cells and that transfection of the miRNA mimic into HepG2 increases miR-122 expression by 57,969 fold. What does this really mean, if the average cell only contains 50,000 miRNAs? These numbers should be admitted as being soft. Firstly, we really need to know the actual, not relative, number of copies of miR-122 per cell. They can be estimated – or if the authors can work quickly, they can get better numbers with small RNAseq. Secondly, as the authors have simply transfected their miRNA mimic into the HepG2 cells, it is likely that the vast majority is actually not loaded into RISC. If they want to really argue for meaningful numbers of the mimic, they would need to do RISC IP, using a pan-Ago antibody, followed by small RNAseq on the pulled down miRNAs. Or back off on the numbers.

3) HepG2 cells were used in most of these studies because ectopic supply of miR-122 to HepG2 cells resulted in a miR-122 abundance that was similar to its abundance in human liver (Figure 1B). However, it is unclear whether miR-122 displays similar roles in innate immunity in other liver cell lines, such as Huh7. Several players in cell growth and differentiation, such as p53, are expressed differently in HepG2 and Huh7 cells. Thus, the authors should repeat a few of the key experiments, such as suppression of STAT3 phosphorylation by miR-122 and inhibition of IRF1 by STAT3, in liver cells other than HepG2.

4) In Figure 1C-F, the authors don't show the results of IFN expression without stimulation. The data of IFN expression without stimulation are required as control. For example, since the difference of IFN-β expression between miR-NC and miR-122 is not huge in Figure 1D, F, the authors should examine if over-expression of miR-122 affects the expression of IFN or not.

5) In Figure 5, the authors use 293FT cells with over-expression of STAT3 for several reporter assays. But revealing the relationship between STAT3 and the expression of IRF1 in hepatocytes is supposed to be the core of the authors' conclusions. The authors should examine the promoter activity of IRF1 in HepG2 cells with or without si-STAT3. A more significant effort would be to examine the promoter activity of P1-M and P4-M in HepG2 cells. This would be helpful but perhaps not essential.

6) In Figure 5F, mutation of STAT3 binding sites within P1 and P4 did not completely abolish the repressing effect of STAT3. And the authors mention the possibility that STAT3 still binds mutant P1 and P4. The authors should perform ChiP experiments to reveal the possibility. In addition, the authors should perform reporter assays with control vectors such as pGL3-control or pGL3-promoter vector in order to examine if STAT3 affects the expression or activity of the reporter gene.

7) In this study, the authors analyze the function of miR-122 by over-expression of miR-122, but they don't perform experiments with inhibiting miR-122. Blocking the function of miR-122 in Huh7 cells and examining of phosphorylation of STAT1/STAT3 or IFN expression would strengthen the authors' conclusions.

References:

Mullokandov G, Baccarini A, Ruzo A, Jayaprakash AD, Tung N, Israelow B, Evans MJ, Sachidanandam R, Brown BD. High-throughput assessment of microRNA activity and function using microRNA sensor and decoy libraries. Nat Methods. 2012 Jul 1;9(8):840-6. doi: 10.1038/nmeth.2078.

---

## [Author Response]

Essential revisions:Our reviewers have a number of criticisms that need to be addressed.1) A concern is that the identification of the mRNA targets for miR-122 has involved some cherry picking. The hit list is long, and only a few are examined further. Other important hits may have been missed, and this should be acknowledged. It would be better to directly identify mRNAs targeted by miR-122 by performing some form of RISC CLIP experiment, such as iCLIP or PAR-CLIP, in the presence and absence of miR-122 using a pan-Ago specific antibody. This will identify all seed target sites for miR-122 occupied in vivo. I would not insist on this last experiment but it would strengthen the paper.

In fact, when we started to screen for the targets that mediate the effect of miR-122 on p-STAT3, we had performed Ago2 PAR-CLIP experiments (Author response image 1). Based on the miR-122-Tet-On HepG2 cell line, we performed PAR-CLIP with two Ago2 antibodies. As summarized in Author response image 1, we totally obtained six CLIP libraries, A1 and M1 were generated from cells with low miR-122 expression (without Dox induction), and the four others were generated from cells with highly abundant miR-122 expression (with Dox induction). As expected, we obtained a number of miR-122-dependent and seed-matched binding sites. However, we found that there are only a few of known STAT3 activators within these candidate miR-122 targets (only 8 genes from the 330 candidate STAT3 activators were detected). Considering that many important targets might be lost in our CLIP data because our CLIP experiments might be not done well enough (we have tried many times before we successfully obtained these six libraries), we decided to screen miR-122 targets based on the microarray data.

Due to the limitation of time, we did not buy a pan-Ago specific antibody and perform PAR-CLIP again. Nowadays, there are many CLIP-seq data have been published and our colleagues have systematically collected and analyzed these data in starBase (http://starbase.sysu.edu.cn/). We find that there are 47 genes from the 330 candidate STAT3 activators have been identified as candidate miR-122 targets in these CLIP-seq data (see Author response table 1). Notably, 9 genes, including FGFR1, IGF1R, IL1RN, ABL2, DSTYK, MAP3K3, EFNA1, IL1R1 and OSMR, which are included in the 20 genes we finally screened out from the microarray data, also can be found in these CLIP-seq data, indicating that our screening method is effective, despite imperfection.

Of course, we admit that our screening method biased towards the known STAT3 activators, some un-identified STAT3 regulators might be lost. So, we have selected 60 candidate miR-122 targets identified by our CLIP experiments and performed an un-biased screening. As shown in Author response image 1, we find that knockdown of several genes could obviously up-regulate p-STAT1 induction and IFN expression in HepG2 cells treated with JFH1 RNA. Knockdown some of them (such as LPAR6, USP34 and STARD7) resulted in a decrease in the level of p-STAT3, though not evident. However, how these genes regulate STAT3 phosphorylation and/or innate immunity is almost unknown, so we would like to perform in-depth research on them in the future.

As our CLIP data may be not good enough to support our paper, we do not provide the data in the revised manuscript. Instead, we provided the data obtained from starBase (Supplementary File 3), and briefly discussed this point in the manuscript (Discussion section), as follows:

“It should be noted that, although our screening method could find out STAT3 regulators that are evidently repressed by miR-122 (their mRNA expression was reduced), the approach might miss some important STAT3 regulators that are only down-regulated at protein level, as miRNAs sometimes only affect the translation of their targets. Through analysis of the candidate miR-122 targets identified by published CLIP-seq data using starBase (Li et al., 2014), we find that there are still dozens of potential STAT3 activators might be direct regulated by miR-122 (Supplementary file 3). Moreover, given that STAT3 could regulate many of its activators in a positive feedback manner (Figure 7—figure supplement 2), the regulation of miR-122 on the STAT3 signaling pathway may be still far from fully understood.”

**Author response table 1. resptable1:** 

Candidate STAT3 activators that predicted to be miR-122 targets in CLIP-seq data
Our CLIP-seq data	Published CLIP-seq data
ABL1, **ABL2**, ERBB3, IFNAR2, IL18BP, JAK1, **MAP3K3**, PRLR	ABL1, **ABL2**, ANGPTL4, CHUK, CSF1, CXCL12, DDR2, **DSTYK, EFNA1**, EFNA3, EFNB2, EPHB2, ERBB3, ERBB4, FGF18, FGF5, **FGFR1**, FGFR3, IFNGR2, **IGF1R**, IGF2R, IL18, IL18BP, **IL1R1**, IL1RL1, **IL1RN**, IL6ST, IL7R, JAK1, KITLG, LIF, MAP3K2, **MAP3K3**, MAPK4, MET, MST1R, NTRK2, **OSMR**, PDGFA, PDGFB, PRLR, RET, ROR1, ROR2, SRC, TEX14, VEGFB

2) The average cell expresses ~50,000 miRNAs or so and miRNA function is dependent on expression level, with miRNAs expressed at <100 copies per cell being essentially non-functional (Mullokandov et al., 2012). Moreover, only RISC associated miRNAs are functional. In Figure 1B, the authors claim that human primary hepatocytes express 81,858 times more miR-122 that HepG2 cells and that transfection of the miRNA mimic into HepG2 increases miR-122 expression by 57,969 fold. What does this really mean, if the average cell only contains 50,000 miRNAs? These numbers should be admitted as being soft. Firstly, we really need to know the actual, not relative, number of copies of miR-122 per cell. They can be estimated – or if the authors can work quickly, they can get better numbers with small RNAseq. Secondly, as the authors have simply transfected their miRNA mimic into the HepG2 cells, it is likely that the vast majority is actually not loaded into RISC. If they want to really argue for meaningful numbers of the mimic, they would need to do RISC IP, using a pan-Ago antibody, followed by small RNAseq on the pulled down miRNAs. Or back off on the numbers.

The numbers shown in the Figure 1B is just to help authors read the figure quickly, as this graph is small, and we used a logarithmic scale in the ordinate axis. Also, we admit that the relative expression level cannot accurately reflect the real copies of miRNA per cell. Due to the limitation of time, we did not measure the number of copies of miR-122 per cell, but choose to remove the numbers in the graph.

3) HepG2 cells were used in most of these studies because ectopic supply of miR-122 to HepG2 cells resulted in a miR-122 abundance that was similar to its abundance in human liver (Figure 1B). However, it is unclear whether miR-122 displays similar roles in innate immunity in other liver cell lines, such as Huh7. Several players in cell growth and differentiation, such as p53, are expressed differently in HepG2 and Huh7 cells. Thus, the authors should repeat a few of the key experiments, such as suppression of STAT3 phosphorylation by miR-122 and inhibition of IRF1 by STAT3, in liver cells other than HepG2.

As suggested, we have tested the roles of miR-122 in other liver cell lines, including Huh7 and Hep3B (Author response image 2). Although miR-122 promoted IFN response in Huh7 cells, but it did not depend on the repression of p-STAT3 (Author response image 2). Similar to the results in HepG2 cells, miR-122 could down-regulate p-STAT3 and up-regulate IFN response in Hep3B cells (Author response image 2). In addition, miR-122 can also down-regulate p-STAT3 in a “normal” mouse liver cells, BNL CL.2 (immortalized mouse fetal liver cell line), suggesting that the regulation of miR-122 on STAT3 phosphorylation is conserved between human and mouse.

Just as mentioned by reviewers, p53 is differently expressed in different liver cancer cell lines. HepG2 has wild-type p53, but Huh7 has mutant p53. Although there is a report showing that p53 represses STAT3 phosphorylation in in human prostate cancer cells (Lin et al., 2002), we find that over-expression of wild-type p53 in Huh7 cells cannot down-regulate STAT3 phosphorylation (Author response image 2). Since miR-122 could down-regulate p-STAT3 in a “p53 null” cell line, Hep3B, we guess that p53 mutation might not be the factor that caused the nonresponse of p-STAT3 to miR-122 in Huh7 cells.

By comparing the protein expression of three RTKs (MERTK, FGFR1 and IGF1R) in HepG2 and Huh7 cells (Author response image 2), we found that all three RTKs are expressed at much lower level in Huh7 cells. Although miR-122 over-expression can down-regulate FGFR1 and IGF1R in Huh7 cells, it cannot lead to a decrease of p-STAT3, suggesting that these RTKs are not responsible for the consistent phosphorylation of STAT3 in Huh7 cells. In consistent with this hypothesis, knockdown of these RTKs (50 nM siRNAs) cannot reduce the p-STAT3 level in Huh7 cells (Author response image 2). Therefore, we surmise that the dysregulation of p-STAT3 in Huh7 might be caused by the up-regulation of some tyrosine kinases that are not directly regulated by miR-122. These STAT3 activators may be not the targets of miR-122. Alternatively, they might be originally regulated by miR-122 in normal hepatocytes, but they have escaped from the control of miR-122 in liver cancer cells, by mutations or by widespread shortening of 3′UTRs (Mayr and Bartel, Cell, 2009).

Although miR-122 did not impact on p-STAT3 in Huh7 cells, it could still slightly up-regulate IFN expression, IRF1 and p-STAT1 (Author response image 2), suggesting that there are backup mechanisms for miR-122 to regulate innate immune pathways. However, comparing the effects of miR-122 overexpression and STAT3 knockdown on IFN response in all tested cell lines confirmed that repressing STAT3 phosphorylation should be the key mechanism by which miR-122 regulates hepatocyte innate immunity.

As suggested, we have added the majority of these data in the revised manuscript (Figure 3—figure supplement 2) and provided a brief discussion as follows:

Results section:

“We also performed assays in Huh7 cells. Unexpectedly, although miR-122 promoted the IFN response, neither over-expression nor blocking of miR-122 affected the expression of p-STAT3 in Huh7 cells (Figure 3—figure supplement 2A and B, Figure 3—source data 8). Nevertheless, knockdown of STAT3 evidently increased the IFN activation in Huh7 cells. To investigate if miR-122 represses p-STAT3 and promotes IFN activation in other liver cells, we further employed Hep3B and BNL CL.2, the former is derived from human HCC cells and the latter is derived from “normal” mouse liver cells (immortalized). Although the p-STAT3 level is relative low in these two cell lines, over-expression of miR-122 obviously reduced p-STAT3 (Figure 3—figure supplement 2C). Notably, miR-122 overexpression and STAT3 knockdown resulted in exactly similar up-regulation of the IFN activation in Hep3B cells (Figure 3—figure supplement 2D, Figure 3—source data 9). These results suggest that miR-122 plays similar roles in these different cells, but the regulatory network from miR-122 to p-STAT3 might has been disrupted in Huh7.”

Discussion section:

“When we investigated miR-122’s roles in different liver cell lines (Figure 3—figure supplement 2), we found that miR-122 promoted the IFN response in Huh7 cells, but did not affect the expression of p-STAT3, suggesting that miR-122 might also regulate innate immunity through other mechanisms. However, comparing the effects of miR-122 overexpression and STAT3 knockdown on IFN responses in all these cell lines (HepG2, Huh7 and Hep3B) confirmed that repressing STAT3 phosphorylation should be the key mechanism by which miR-122 regulates innate immunity. Given that these cell lines are derived from HCC patients with different backgrounds, further understanding on the differences observed in these cells would shed new light on the roles of both miR-122 and STAT3 in liver biology and disease.”

**Author response image 2. respfig2:** 

4) In Figure 1C-F, the authors don't show the results of IFN expression without stimulation. The data of IFN expression without stimulation are required as control. For example, since the difference of IFN-β expression between miR-NC and miR-122 is not huge in Figure 1D, F, the authors should examine if over-expression of miR-122 affects the expression of IFN or not.

We had performed this experiment before. The expression of IFNs is very low in cells without stimulation. IFN mRNAs can be detected (by qRT-PCR), but IFN proteins cannot be detected (by ELISA). Additionally, miR-122 overexpression did not impact on the IFN expression in cells without stimulation.

As suggested, we have added the qPCR data into Figure 1C and added a sentence in the results “Surprisingly, miR-122 overexpression markedly increased the IFN induction in response to almost all stimuli we tested, but did not impact on the IFN expression in cells without stimulation.”

5) In Figure 5, the authors use 293FT cells with over-expression of STAT3 for several reporter assays. But revealing the relationship between STAT3 and the expression of IRF1 in hepatocytes is supposed to be the core of the authors' conclusions. The authors should examine the promoter activity of IRF1 in HepG2 cells with or without si-STAT3. A more significant effort would be to examine the promoter activity of P1-M and P4-M in HepG2 cells. This would be helpful but perhaps not essential.

In fact, we had performed STAT3 knockdown experiments in HepG2 cells before we performed reporter assays in 293FT cells. However, we did not show the data in the results, because we felt that the results were not significant and we could not explain why. At that time, we used the pGL3-basic vector as the control (its activity did not alter after STAT3 knockdown), and we found that STAT3 knockdown only slightly up-regulate the activity of P7 reporter. This time, as suggested, we performed the same experiments using the pGL3-control vector as the control. Unexpectedly, we find that the activity of pGL3-control is significantly down-regulated (30%-40%) in HepG2 cells when STAT3 is knockdown (Author response image 3). By comparing the activities of P1, P7 and pGL3-control, we find that P1 and P7 promoters are very active in HepG2 cells, and the relative activities of them can reach to approximately 30%-80% of pGL3-control activity (Author response image 3). So, it seems that STAT3 knockdown could impact on the activities of strong promoters in a non-specific manner. After normalization to the activity of pGL3-control, we found that the effect of STAT3 knockdown on all three promoters (P7, P1 and P4) is highly significant (Author response image 3).

As suggested, we have added these data into the results (Figure 5D and G).

**Author response image 3. respfig3:** 

6) In Figure 5F, mutation of STAT3 binding sites within P1 and P4 did not completely abolish the repressing effect of STAT3. And the authors mention the possibility that STAT3 still binds mutant P1 and P4. The authors should perform ChiP experiments to reveal the possibility. In addition, the authors should perform reporter assays with control vectors such as pGL3-control or pGL3-promoter vector in order to examine if STAT3 affects the expression or activity of the reporter gene.

As suggested, we have performed ChIP assays on mutant P1 and P4 (Author response image 4). We transfected wild-type or mutant reporter constructs into HepG2 cells, and then performed STAT3 ChIP assays with PCR primers that can specifically detect the IRF1 promoters on the reporter constructs. As expected, the mutation (P1-M and P4-M) obviously reduced the binding of STAT3 on these two promoters. However, STAT3 still bind mutant promoters, especially of P4.

Also, we have repeated reporter assays in both HepG2 and 293FT cells using the pGL3-control vector as a control. Just as our reply to the Point 5, knockdown of STAT3 unexpectedly reduced the activity of pGL3-control vector (30%-40%) in HepG2 cells (Author response image 4). But in 293FT cells, over-expression of STAT3 did not significantly impact on the activity of pGL3-control vector (Author response image 4). Here, we thank the reviewers for their suggestion on using pGL3-control as a control, which helped us understand the results we could not explain before.

As suggested, we have added these data into the results in the revised manuscript (Figure 5D, E and I).

**Author response image 4. respfig4:** 

7) In this study, the authors analyze the function of miR-122 by over-expression of miR-122, but they don't perform experiments with inhibiting miR-122. Blocking the function of miR-122 in Huh7 cells and examining of phosphorylation of STAT1/STAT3 or IFN expression would strengthen the authors' conclusions.

As suggested, we have performed miR-122 inhibition experiments in Huh7 cells. Although blocking miR-122 with antisense inhibitors can slightly up-regulate FGFR1 and IGF1R, it did not up-regulate p-STAT3 (Author response image 5). This result is consistent with our previous observation that over-expression of these two RTKs only slightly increased STAT3 phosphorylation in Huh7 cells (Figure 6F). Nevertheless, inhibiting of miR-122 decreased the expression of IRF1 (Author response image 5) and slightly down-regulated the induction of IFNs (Author response image 5).

As suggested, we have added these data into the supplementary data in the revised manuscript (Figure 3—figure supplement 2A), in combination with other data obtained in Huh7 cells.

**Author response image 5. respfig5:** 

References:

Lin J, Tang H, Jin X, Jia G, Hsieh JT. p53 regulates Stat3 phosphorylation and DNA binding activity in human prostate cancer cells expressing constitutively active Stat3. Oncogene. 2002 May 2;21(19):3082-8.

Mayr C, Bartel DP. Widespread Shortening of 3 ' UTRs by Alternative Cleavage and Polyadenylation Activates Oncogenes in Cancer Cells. Cell. 2009 Aug 21;138(4):673-84.